# What type of inference is planning?

**Miguel Lázaro-Gredilla**    **Li Yang Ku**    **Kevin P. Murphy**    **Dileep George**
Google Deepmind
{lazarogredilla, liyangku, kpmurphy, dileepgeorge}@google.com

## Abstract

Multiple types of inference are available for probabilistic graphical models, e.g., marginal, maximum-a-posteriori, and even marginal maximum-a-posteriori. Which one do researchers mean when they talk about "planning as inference"? There is no consistency in the literature, different types are used, and their ability to do planning is further entangled with specific approximations or additional constraints. In this work we use the variational framework to show that, just like all commonly used types of inference correspond to different weightings of the entropy terms in the variational problem, planning corresponds *exactly* to a *different* set of weights. This means that all the tricks of variational inference are readily applicable to planning. We develop an analogue of loopy belief propagation that allows us to perform approximate planning in factored-state Markov decisions processes without incurring intractability due to the exponentially large state space. The variational perspective shows that the previous types of inference for planning are only adequate in environments with low stochasticity, and allows us to characterize each type by its own merits, disentangling the type of inference from the additional approximations that its practical use requires. We validate these results empirically on synthetic MDPs and tasks posed in the International Planning Competition.

## 1 Introduction

There are many kinds of probabilistic inference, such as marginal, maximum-a-posteriori (MAP), or marginal MAP (MMAP) that are used in the planning as inference literature (Attias, 2003; Levine, 2018; Cui et al., 2019; Palmieri et al., 2022; Wu and Khardon, 2022). In this work we show that planning is a distinct type of inference, and that *under stochastic dynamics* does not correspond exactly with any of the above methods. Furthermore, we show how to rank the above methods in terms of quality as it pertains to planning.

Our approach is based on a variational perspective, which allows direct comparison between different inference types, and to develop analogues of existing approximate inference algorithms for this "planning inference" task. Given the "flat" Markov Decision Process (MDP) from Fig 1[Left], we show that planning inference provides the same (exact) results as value iteration. Using an analogue of loopy belief propagation (LBP), we show how to apply approximate planning inference to the factored MDP in Fig. 1[Right], which has an exponentially large state space, and for which exact solutions are no longer tractable. Under moderate stochasticity in the dynamics, we show that this approximate planning inference might be superior to other more established types of inference.

## 2 Background

### 2.1 Markov Decision processes (MDPs) and notation

A finite-horizon Markov decision process (MDP) is a tuple $(\mathcal{X}, \mathcal{A}, p(x_1), \mathcal{P}, \mathcal{R}, T)$, where $\mathcal{X}$ is the state space, $\mathcal{A}$ an action space with cardinality $N_a$, $p(x_1)$ the starting state distribution, $\mathcal{P}$ the

38th Conference on Neural Information Processing Systems (NeurIPS 2024).

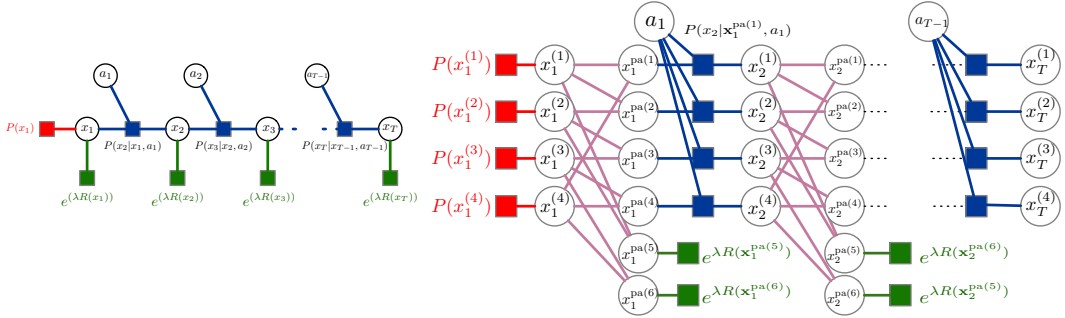

Figure 1: Factor graphs: [Left] Standard MDP [Right] Factored MDP with sparse factor connectivity.

transition probabilities $P(x_{t+1}|a_t, x_t)$ (i.e., the dynamics of the process), $R_t(x_t, a_t, x_{t+1})$ the reward for transitioning from $x_t$ to $x_{t+1}$ under action $a_t$ at time step $t$, and $T$ is the horizon. For simplicity of exposition we will consider discrete states and actions, but analogous results apply in the continuous case. Solving an MDP corresponds to finding a policy $\pi_t(a_t|x_t)$ that maximizes the expectation of (a function of) the sum of the reward at each time step. The optimal policy can be different at each time step, i.e., non-stationary. We do not use a discount factor, but it can be trivially included in the reward function, since it is also non-stationary. Policies, states and actions at all time steps will be noted $\boldsymbol{\pi} \equiv \{\pi_t(a_t|x_t)\}_{t=1}^{T-1}, \boldsymbol{x} \equiv \{x_t\}_{t=1}^{T}$ and $\boldsymbol{a} \equiv \{a_t\}_{t=1}^{T-1}$.

In a state-factored MDP (factored MDP for short), each state $x_t$ factorizes into $N_e$ r.v. or *entities* $\{x_t^{(i)}\}_{i=1}^{N_e}$, each with cardinality $N_s$, so it has an exponentially large state space of size $N_s^{N_e}$. Transitions factorize as $P(x_{t+1}|a_t, x_t) = \prod_{i=1}^{N_e} P(x_{t+1}^{(i)}|x_t^{\mathrm{pa}(i)}, a_t)$, where $x_t^{\mathrm{pa}(i)}$ is the subset of $x_t$ on which $x_{t+1}^{(i)}$ depends, which we will assume to be small to allow for a tabular definition of the transition without exponential cost (pa($i$) stands in for "parents of $i$"). For simplicity of notation, we will assume that the reward of factored MDPs at each time step depends only on the current state, and for tractability, that it can be decomposed in multiple additive subterms $R_t(x_t) = \sum_{i=N_e+1}^{N_e+N_r} R_t(x_t^{\mathrm{pa}(i)})$, where $x_t^{\mathrm{pa}(i)}$ is a small subset of $x_t$ on which the $(i - N_e)$-th reward depends, for a total of $N_r$ reward subterms. As before, this allows for a compact tabular representation of the reward. Including additional dependencies in the reward (on actions, or on the next state of an entity) is straightforward.

For a more comprehensive introduction to MDPs, refer to (Puterman, 2014; Sutton, 2018).

## 2.2 Variational inference

Let's consider running different *types* of inference in the unnormalized[1] factor graph shown in Fig. 1[Left]. Marginal inference would compute the sum over all the $\boldsymbol{x}, \boldsymbol{a}$ configurations (the partition function); MAP inference would compute the maximizing configuration (the posterior mode); and marginal MAP (MMAP) inference (see Liu and Ihler, 2013) would find the assignment of $\boldsymbol{a}$ that maximizes the summation over $\boldsymbol{x}$ conditional on that $\boldsymbol{a}$. Although marginal, MAP, and MMAP inference are distinct, a lot is shared. They all target a "quantity of interest" (e.g., partition function, maximum probability state, best conditional partition function); they all produce a distribution as a result of inference (respectively, full posterior, delta at the mode, best conditional posterior); naïve computation requires a number of summations and/or maximizations exponential in the number of variables; and finally, they can all be represented as a variational inference (VI) problem.[2] The VI representation naturally leads to efficient message-passing solutions and approximate inference algorithms, as we show below.

For the factor graph $f(\boldsymbol{x}, \boldsymbol{a})$ from Fig. 1[Left], the VI problem[3] is $\max_{q(\boldsymbol{x},\boldsymbol{a})} \langle \log f(\boldsymbol{x}, \boldsymbol{a}) \rangle_{q(\boldsymbol{x},\boldsymbol{a})} + H_q^{\mathrm{type}}(\boldsymbol{x}, \boldsymbol{a})$ where $q(\boldsymbol{x}, \boldsymbol{a})$ is an arbitrary variational distribution over the variables of the factor

---

[1]Just like in undirected graphical models, the different types of inference are well-defined despite lack of normalization. Table 1 shows the precise definitions including two reasonable definitions of marginalization.

[2]We assume familiarity with variational inference, see (Jordan et al., 1999) for an introduction.

[3]In this work we will use the standard VI notation for expectation, where $\mathbb{E}_{q(x)}[\cdot]$ is written as $\langle \cdot \rangle_{q(x)}$.

Table 1: Different types of inference from a variational perspective, including a proper "planning" inference type. They all share the same energy term $E_\lambda(\boldsymbol{q})$ defined in Eq. (3), and differ only in the entropy term. The closed-form expressions provide the optimal value of the bound, but are generally intractable. The general tractability of the bound maximization for MDPs is marked in the Tr column. All bounds are monotonically related, and listed in descending order, except the last two, whose relative ordering only applies when dynamics are deterministic. See Section 4.1 for details.

| Type of inference $\downarrow$ | Closed form for quant. of interest $\downarrow$ $F_\lambda = \max_{\boldsymbol{q}} F_\lambda(\boldsymbol{q})$ | Entropy term $H^{\text{type}}(\boldsymbol{q})$ for variational bound $\downarrow$ $F_\lambda(\boldsymbol{q}) = \frac{1}{\lambda}(-E_\lambda(\boldsymbol{q}) + H^{\text{type}}(\boldsymbol{q}))$ | Tr |
|---|---|---|---|
| Marginal[4] | $\frac{1}{\lambda} \log \sum_{\boldsymbol{x},\boldsymbol{a}} P(\boldsymbol{x}|\boldsymbol{a})e^{\lambda R(\boldsymbol{x},\boldsymbol{a})}$ | $H_q(x_1) + \sum_{t=1}^{T-1} H_q(x_{t+1}, a_t|x_t)$ | ✓ |
| **Planning** | $\frac{1}{\lambda} \max_{\boldsymbol{\pi}} \log \langle e^{\lambda R(\boldsymbol{x},\boldsymbol{a})} \rangle_{P(\boldsymbol{x}|\boldsymbol{a})\pi(\boldsymbol{a}|\boldsymbol{x})}$ | $H_q(x_1) + \sum_{t=1}^{T-1} H_q(x_{t+1}|a_t, x_t)$ | ✓ |
| M. MAP | $\frac{1}{\lambda} \max_{\boldsymbol{a}} \log \sum_{\boldsymbol{x}} P(\boldsymbol{x}|\boldsymbol{a})e^{\lambda R(\boldsymbol{x},\boldsymbol{a})}$ | $H_q(x_1) + \sum_{t=1}^{T-1} H_q(x_{t+1}, a_t|x_t) - H_q(a_t)$ | ✗ |
| MAP | $\frac{1}{\lambda} \max_{\boldsymbol{x},\boldsymbol{a}} \log P(\boldsymbol{x}|\boldsymbol{a})e^{\lambda R(\boldsymbol{x},\boldsymbol{a})}$ | $0$ | ✓ |
| Marginal[U] | $\frac{1}{\lambda} \log \sum_{\boldsymbol{x},\boldsymbol{a}} P(\boldsymbol{x}|\boldsymbol{a})\frac{1}{N_a^{T-1}} e^{\lambda R(\boldsymbol{x},\boldsymbol{a})}$ | $H_q(x_1) + \sum_{t=1}^{T-1} (H_q(x_{t+1}, a_t|x_t) - \log N_a)$ | ✓ |

graph, the term $-\langle \log f(\boldsymbol{x}, \boldsymbol{a}) \rangle_{q(\boldsymbol{x},\boldsymbol{a})}$ is known as the *energy term*, and $H_q^{\text{type}}(\boldsymbol{x}, \boldsymbol{a})$ is a particular entropy choice that will determine the inference type. It is a standard result, (e.g., Jordan et al., 1999) that using the Shannon entropy $H_q(\boldsymbol{x}, \boldsymbol{a})$ results in marginal inference. Setting it to zero (aka the zero-temperature limit) corresponds to MAP inference (Weiss et al., 2012; Sontag et al., 2011; Wainwright et al., 2005; Kolmogorov, 2005; Martins et al., 2015). Setting it to the conditional entropy $H_q(\boldsymbol{x}|\boldsymbol{a}) = H_q(\boldsymbol{x}, \boldsymbol{a}) - H_q(\boldsymbol{a})$ results in MMAP inference (Liu and Ihler, 2013). Note that the only difference across VI problems is the *weighting* of the entropy terms.

Despite the similarities, the computational complexity of these types of inference can differ significantly, even for tree-structured graphs. In particular, the entropy term for MMAP is not concave, and inference is NP-hard (Liu and Ihler, 2013), whereas for marginal and MAP inference the entropy term is concave (the energy term is always linear), and inference is polynomial.

# 3 Methods

In this section we introduce our VI framework, which we will use to derive a novel linear programming formulation of planning problems, a novel value belief propagation (VBP) algorithm and a novel closed form (sampling-free) approach to determinization.

## 3.1 VI for standard MDPs

The main quantity of interest in this paper is the *best exponential utility*, which we will refer to simply as the *utility*. Given an MDP with horizon $T$ and a risk parameter $\lambda > 0$, the utility is defined as

$$F_\lambda^{\text{planning}} = \frac{1}{\lambda} \log \max_{\boldsymbol{\pi}} \sum_{\boldsymbol{x},\boldsymbol{a}} \exp\left(\lambda \sum_{t=1}^{T-1} R_t(x_t, a_t, x_{t+1})\right) P(x_1) \prod_{t=1}^{T-1} P(x_{t+1}|a_t, x_t)\pi_t(a_t|x_t) \quad (1)$$

$$= \frac{1}{\lambda} \max_{\boldsymbol{\pi}} \log \langle \exp(\lambda R(\boldsymbol{x}, \boldsymbol{a})) \rangle_{P(\boldsymbol{x}|\boldsymbol{a})\pi(\boldsymbol{a}|\boldsymbol{x})},$$

where $P(\boldsymbol{x}|\boldsymbol{a}) \equiv P(x_1) \prod_{t=1}^{T-1} P(x_{t+1}|a_t, x_t)$, $R(\boldsymbol{x}, \boldsymbol{a}) \equiv \sum_{t=1}^{T-1} R_t(x_t, a_t, x_{t+1})$, and $\pi(\boldsymbol{a}|\boldsymbol{x}) \equiv \prod_{t=1}^{T-1} \pi_t(a_t|x_t)$.

Observe that we can always set $\lambda \to 0^+$ to recover the standard planning setting in which we seek the best expected *additive reward*, so here we are tackling a strictly more general case. To be precise, if we take the limit $F_{\lambda \to 0^+}^{\text{planning}} \equiv \lim_{\lambda \to 0^+} F_\lambda^{\text{planning}} = \max_{\boldsymbol{\pi}} \langle R(\boldsymbol{x}, \boldsymbol{a}) \rangle_{P(\boldsymbol{x}|\boldsymbol{a})\pi(\boldsymbol{a}|\boldsymbol{x})}$ (Marthe et al., 2023).

The motivation for the introduction of $\lambda$ is two-fold. On the one hand, by using a more general formulation of the reward, we can trade off between risk-neutral ($\lambda \to 0^+$) and risk-seeking ($\lambda > 0$)

---

[4]Two types of marginal inference are included for precision. "Marginal" refers to marginal inference directly applied on the same exact factor graph as the other types of inference. Because the factor graph lacks a prior over $\boldsymbol{a}$, it is not a properly normalized joint distribution. Adding a uniform prior over the actions resolves this and results in "Marginal[U]". Both are a constant apart and are of independent interest.

policies, adding a tunable parameter that makes the model more flexible, see (Marthe et al., 2023; Föllmer and Schied, 2011; Shen et al., 2014) for more details. On the other hand, it allows us to express the expected reward as a proper factor graph: note that Eq. (1) can be expressed as a product of factors involving not only the dynamics terms, but also the reward terms, allowing us to write it as $F_\lambda^{\text{planning}} = \frac{1}{\lambda} \log \max_{\boldsymbol{\pi}} \sum_{\boldsymbol{x}, \boldsymbol{a}} f(\boldsymbol{x}, \boldsymbol{a}) \pi(\boldsymbol{a}|\boldsymbol{x})$, where $f(\boldsymbol{x}, \boldsymbol{a})$ is the factor graph of Fig. 1[Left]. This factorization would not have been possible if we had simply used an additive reward. But at the same time, notice that we are not losing generality, since the additive reward case can be recovered by setting $\lambda \to 0^+$. Alternatively, we could have achieved a factorized model by introducing an additional latent "selector" variable connected to all the rewards, but this would complicate our upcoming formulation and analysis. Furthermore, this formulation allows us to encompass prior work on planning as inference that uses $\lambda > 0$.

We can turn this new quantity of interest, the utility, into the solution of a VI problem on the factor graph of Fig. 1[Left]. Crucially, the factor graph includes the known dynamics and rewards terms, but *not* any policy term, since the policy is the outcome of inference.

**Theorem 1** (Variational formulation of planning). *Given known dynamics $P(x_{t+1}|a_t, x_t)$, an initial distribution $P(x_1)$ and reward functions $R_t(x_t, a_t, x_{t+1})$, the best exponential utility $F_\lambda^{planning}$ from Eq. (1) can be expressed as the result of a concave variational optimization problem*

$$F_\lambda^{planning} = \max_{\boldsymbol{q}} F_\lambda^{planning}(\boldsymbol{q}); \qquad F_\lambda^{planning}(\boldsymbol{q}) = \frac{1}{\lambda}(-E_\lambda(\boldsymbol{q}) + H^{planning}(\boldsymbol{q})) \qquad (2)$$

*with energy $E_\lambda(\boldsymbol{q})$ and entropy $H^{planning}(\boldsymbol{q})$ terms*

$$E_\lambda(\boldsymbol{q}) = -\langle \log P(x_1) \rangle_{q(x_1)} - \sum_{t=1}^{T-1} \langle \log P(x_{t+1}|a_t, x_t) + \lambda R_t(x_t, a_t, x_{t+1}) \rangle_{q(x_{t+1}, x_t, a_t)} \qquad (3)$$

$$H^{planning}(\boldsymbol{q}) = H_q(x_1) + \sum_{t=1}^{T-1} H_q(x_{t+1}|a_t, x_t) \qquad (4)$$

*where $\boldsymbol{q} \equiv q(\boldsymbol{x}, \boldsymbol{a})$ is an arbitrary distribution over the space of states and actions.*

Proof is in Appendix A. This entropy thus corresponds to "planning inference". The optimal policy at each time step corresponds to the optimal variational distribution $q(a_t|x_t)$. Table 1 lists the types of inference problems and their associated entropies (see Appendix F for their derivation and corresponding references). As we will discuss in Section 4, they display a monotonic ordering (in almost all cases).

The VI problem Eq. (2) reduces to the standard one when $\lambda = 1$, and extends VI in a meaningful way in the presence of rewards, regardless of the type of inference used: rewards interact in an additive way when $\lambda \to 0^+$, rather than the default multiplicative (or more precisely, exponentiated summation) interaction of $\lambda = 1$. Furthermore, it turns out that it is possible to take the $\lambda \to 0^+$ limit exactly, to obtain the dual LP formulation of an MDP (Puterman, 2014).

**Corollary 1.1** (Additive limit). *In the limit $\lambda \to 0^+$, the concave problem Eq. (2) becomes the following linear program (LP):*

$$F_{\lambda \to 0^+}^{planning} = \max_{\boldsymbol{q}} F_{\lambda \to 0^+}^{planning}(\boldsymbol{q}) = \max_{\{q(x_t, a_t)\}_{t=1}^{T-1}} \sum_{t=1}^{T-1} \langle R_t(x_t, a_t, x_{t+1}) \rangle_{P(x_{t+1}|a_t, x_t) q(x_t, a_t)}$$

$$s.t. \ q(x_1) = P(x_1); \qquad \sum_{a_{t+1}} q(x_{t+1}, a_{t+1}) = \sum_{x_t, a_t} P(x_{t+1}|a_t, x_t) q(x_t, a_t) \ \forall t;$$

$$q(x_t) = \sum_{a_t} q(x_t, a_t) \ \forall t; \qquad q(x_t, a_t) \geq 0 \ \forall t,$$

*which corresponds to the maximum expected reward $F_{\lambda \to 0^+}^{planning} = \max_{\boldsymbol{\pi}} \langle R(\boldsymbol{x}, \boldsymbol{a}) \rangle_{P(\boldsymbol{x}|\boldsymbol{a}) \pi(\boldsymbol{a}|\boldsymbol{x})}$.*

See Appendix B for proof.

## 3.2 VI LP and VBP for factored MDPs

Factored MDPs (e.g., Fig. 1[Right]) are loopy factor graphs with an exponentially large state space, so the previous approaches cannot be applied directly. An effective approximate marginal inference approach for this type of problem is loopy belief propagation (LBP). Since planning is now seen as a type of inference, we can create an analogue to LBP which we call value belief propagation (VBP).

Following LBP, we make two approximations to Eq. (2) to make it tractable.

First, we replace the variational distribution $q$ with pseudo-marginals $\tilde{q}$. Eqs. (3) and (4) never access the full joint $q(\boldsymbol{x}, \boldsymbol{a})$, but only the local marginals of each factor. Pseudo-marginals are the collection of such local distributions, consistent at each variable, but not necessarily marginals of any distribution. Just like $q$ is defined in a convex region called the *marginal polytope* $\mathcal{M}$, $\tilde{q}$ is defined in an outer convex region called the *local polytope* $\mathcal{L}$ that contains $\mathcal{M}$ (Weller et al., 2014).

Second, we replace the entropy $H^{\text{planning}}(\boldsymbol{q})$ with its *Bethe* approximation

$$H_{\text{Bethe}}^{\text{planning}}(\tilde{\boldsymbol{q}}) = \sum_{i=1}^{N_e} H_q(x_1^{(i)}) + \sum_{t=1}^{T-1} \Big( \sum_{i=1}^{N_e} H_q(x_{t+1}^{(i)}|x_t^{\text{pa}(i)}, a_t) - \sum_{i=1}^{N_e+N_r} I_q(x_t^{\text{pa}(i)}) \Big)$$

where $I_q(x_t^{\text{pa}(i)}) = \sum_{k \in \text{pa}(i)} H_q(x_t^{(k)}) - H_q(x_t^{\text{pa}(i)})$ is the mutual information of the parents of $x_{t+1}^{(i)}$, see Appendix C for details. This approximation is tractable as long as the factors (transition and rewards) are tractable, typically by connecting to a small number of parent variables. Note that the policy (which would connect to all the state variables in a time slice and introduce exponential cost) is not a factor in the graph.

The term $I_q(x_t^{\text{pa}(i)})$ is key. It always non-negative but neither concave nor convex in general and can be interpreted as the mutual information correcting the discrepancy between (a) the entropy of a collection of variables considered independently (as the output of the previous time step) and (b) the entropy of the same collection when considered jointly (as parents for the current time step). It makes the optimization problem harder, but also more accurate.

For non-factored MDPs, $I_q(x_t^{\text{pa}(i)}) = 0$ and $\tilde{\boldsymbol{q}} = \boldsymbol{q}$, so we recover Eq. (2). In general factored MDPs this is not true. We can still choose to ignore this correction to obtain a concave bound

$$H_{\text{concave}}^{\text{planning}}(\tilde{\boldsymbol{q}}) = \sum_{i=1}^{N_e} H_q(x_1^{(i)}) + \sum_{t=1}^{T-1} \sum_{i=1}^{N_e} H_q(x_{t+1}^{(i)}|x_t^{\text{pa}(i)}, a_t) \geq H_{\text{Bethe}}^{\text{planning}}(\tilde{\boldsymbol{q}}). \tag{5}$$

Then, the planning Bethe approximation of the variational bound and its concave upper bound are

$$\tilde{F}_\lambda^{\text{planning}} = \max_q \tilde{F}_\lambda^{\text{planning}}(\tilde{\boldsymbol{q}}) = \max_q \frac{1}{\lambda}(-E_\lambda(\tilde{\boldsymbol{q}}) + H_{\text{Bethe}}^{\text{planning}}(\tilde{\boldsymbol{q}})) \quad s.t. \ \tilde{\boldsymbol{q}} \in \mathcal{L} \tag{6}$$

$$\hat{F}_\lambda^{\text{planning}} = \max_q \hat{F}_\lambda^{\text{planning}}(\tilde{\boldsymbol{q}}) = \max_q \frac{1}{\lambda}(-E_\lambda(\tilde{\boldsymbol{q}}) + H_{\text{concave}}^{\text{planning}}(\tilde{\boldsymbol{q}})) \quad s.t. \ \tilde{\boldsymbol{q}} \in \mathcal{L}. \tag{7}$$

We see that $\hat{F}_\lambda^{\text{planning}} \geq \tilde{F}_\lambda^{\text{planning}}$ and $\hat{F}_\lambda^{\text{planning}} \geq F_\lambda^{\text{planning}}$. The former is trivial given the negative term removed. The latter follows from (a) switching the optimization domain from $\mathcal{M}$ to $\mathcal{L} \supseteq \mathcal{M}$, which can only increase the value of the bound, and (b) Eq. (5) corresponds to Eq. (4), but with joint entropies over entities replaced with sums of the entropies, which is an upper bound. The fact that $\hat{F}_\lambda^{\text{planning}}(\tilde{\boldsymbol{q}})$ is concave and upper bounds the exact utility has two advantages: it can be computed without local minima problems, and it is an *admissible heuristic* of the original utility, meaning that it can be used as a heuristic for algorithms that emit a certificate of optimality or infeasibility.

**Lemma** (Additive limit for factored MDPs). *In the limit $\lambda \to 0^+$, the concave problem Eq. (7) becomes the following VI LP:*

$$\hat{F}_{\lambda \to 0^+}^{planning} = \max_{\tilde{\boldsymbol{q}}} \hat{F}_{\lambda \to 0^+}^{planning}(\tilde{\boldsymbol{q}}) = \max_{\tilde{\boldsymbol{q}}} \sum_{t=1}^{T} \sum_{i=N_e}^{N_e+N_r} \langle R_t(x_t^{pa(i)}) \rangle_{q(x_t^{pa(i)})}$$

$$s.t. \quad q(x_1^{(i)}) = P(x_1^{(i)}) \ \forall i; \quad \tilde{\boldsymbol{q}} \in \mathcal{L}$$

$$q(x_{t+1}^{(i)}) = \sum_{x_t^{pa(i)}, a_t} P(x_{t+1}^{(i)}|x_t^{pa(i)}, a_t) q(x_t^{pa(i)}, a_t) \ \forall x_{t+1}^{(i)}, t, 1 \leq i \leq N_e$$

*which upper bounds the max. expected reward $\hat{F}_{\lambda \to 0^+}^{planning} \geq F_{\lambda \to 0^+}^{planning} = \max_{\boldsymbol{\pi}} \langle R(\boldsymbol{x}, \boldsymbol{a}) \rangle_{P(\boldsymbol{x}|\boldsymbol{a})\pi(\boldsymbol{a}|\boldsymbol{x})}$. Alternatively, the same expression can be obtained from Corollary 1.1 by relaxing the marginal polytope into the local polytope. Since it is a relaxation, the upper bounding is trivial.*

To the best of our knowledge, this is a novel VI LP formulation and it can be used to tractably (over) estimate the optimal expected reward in factored MDPs. Similarities with (Koller and Parr, 1999; Guestrin et al., 2003) are only surface level, see Section 5.

$\hat{F}_{\lambda}^{\text{planning}}(\tilde{\boldsymbol{q}})$ from Eq. (7) can be maximized with a conic solver (or an LP solver if $\lambda \to 0^+$). The non-concave $\tilde{F}_{\lambda}^{\text{planning}}(\tilde{\boldsymbol{q}})$ from Eq. (6) is more challenging. Conveniently, $\tilde{F}_{\lambda}^{\text{planning}}(\tilde{\boldsymbol{q}})$ looks just like the Bethe free energy that motivates LBP, but with a different weighting of the local entropy terms.

Multiple works consider modifying the entropy weighting in LBP, usually with the aim of "concavifying" the overall entropy term and developing convergent alternatives to LBP. In particular, (Hazan and Shashua, 2010) provide fixed-point message updates for arbitrary entropy weights. For the specific weighting of $H_{\text{Bethe}}^{\text{planning}}(\tilde{\boldsymbol{q}})$ the message updates approach a singularity, which we will avoid by using $(1-\epsilon)H_{\text{Bethe}}^{\text{planning}}(\tilde{\boldsymbol{q}})) + \epsilon H_{\text{Bethe}}^{\text{marginal}}(\tilde{\boldsymbol{q}})$. The resulting message passing algorithm updates are well-defined for any $\epsilon > 0$, interpolate between "planning inference" and marginal inference, and can get arbitrarily close to the former by making $\epsilon \to 0^+$. This smoothing is not just a mathematical convenience, but we prove that it exactly corresponds to MaxEnt RL in Appendix E. See (Liu and Ihler, 2013) for an analogous technique with the same purpose (but without this nice interpretation).

VBP inherits many of the properties of LBP: the message updates are not guaranteed to converge, but if they do, they do so at a fixed point of Eq. (6). Convergence can be improved by the use of damping and annealing. The precise message updates for the general case are provided in Appendix D.

Computation associated with VBP scales as expected, $\mathcal{O}(T(\sum_{i=1}^{N_e} N_a N_s^{|\text{pa(i)}|+1} + \sum_{i=N_e+1}^{N_r} N_s^{|\text{pa(i)}|}))$, where $N_e, N_r, N_a, N_s$ have been defined in Section 2.1. Note that the derivation is straightforward. Each VBP iteration involves computing message updates for each factor in the graph. The cost is dominated by the blue factors ($N_e$ of them per time step) and green factors ($N_r$ of them per time step) in Fig. 1[Right]. There are a total of $T$ time steps. And finally the number of possible configurations is $N_a N_s^{|\text{pa(i)}|+1}$ for blue factors and $N_s^{|\text{pa(i)}|}$ for green factors.

### 3.3 VBP for standard MDPs

It is instructive to look at the VBP updates for a standard, non-factored MDP. In this case, it is possible to take the limit $\epsilon \to 0^+$ and get well-defined updates. For $\lambda = 1$ and a single reward at $T$

Backward updates: $m_{\text{b}}(x_T) = e^{R_T(x_T)}; \quad m_{\text{b}}(x_t) = \max_{a_t} Q(x_t, a_t);$

Forward updates: $m_{\text{f}}(x_1) = P(x_1); \quad m_{\text{f}}(x_{t+1}) = \sum_{x_t, a_t} p(x_{t+1}|x_t, a_t)\delta_{a_t, \text{argmax}_{a'_t} Q(x_t, a'_t)} m_{\text{f}}(x_t)$

Optimal dist.: $q(x_{t+1}, x_t, a_t) \propto m_{\text{b}}(x_{t+1})p(x_{t+1}|x_t, a_t)\delta_{a_t, \text{argmax}_{a'_t} Q(x_t, a'_t)} m_{\text{f}}(x_t)$

where $Q(x_t, a_t) = \sum_{x_{t+1}} m_{\text{b}}(x_{t+1})p(x_{t+1}|x_t, a_t)$ and $\delta_{j,k}$ is a standard Kronecker delta that equals 1 when $j = k$ and 0 otherwise. Iterating these updates converges in a single backward and forward pass to the global optimum. The backward messages correspond to the value function (hence the name VBP), and the familiar intermediate quantity $Q(x_t, a_t)$ matches the Q-function. The forward messages correspond to occupancy probabilities under the optimal policy. Thus, in a non-factored MDP we recover the standard Bellman backups, implementing value iteration and providing the exact solution. The same happens, conceptually, in a factored MDP, but only approximately, with the forward messages helping to determine where the backward approximation should be more precise.

### 3.4 Determinization in hindsight

The previous presentation implies that all VI tricks are now applicable to planning. As an example, we can show that for *determinization* (Yoon et al., 2008) (a technique from the planning literature to extend deterministic planning algorithms to stochastic domains, and that is usually computed via sampling), we can obtain a precise upper bound as the solution of a tractable concave problem.

To be more precise, we can compute $F_{\lambda=1}^{\text{det. planning UB}}$ (for MDPs) and $\hat{F}_{\lambda=1}^{\text{det. planning UB}}$ (for factored MDPs) as a concave optimization problem (avoiding sampling) when the inner deterministic planning problem is solved with an LP MAP relaxation (exact for MDPs). Additionally, we can prove that for factored MDPs $\hat{F}_{\lambda=1}^{\text{planning}} \leq \hat{F}_{\lambda=1}^{\text{det. planning UB}}$ (i.e., the superiority of the bound introduced here wrt this determinization upper bound in the case of factored MDPs). See Appendix H for further details.

## 4 The different types of inference and their adequacy for planning

### 4.1 Ranking inference types for planning

As we show in Section 5, the term "planning as inference" has been used in the literature to refer to different inference types, none of which corresponds, to the best of our knowledge, with the "planning inference" from this work, which is exact. Table 1 associates each type of inference to a corresponding lower bound on its quantity of interest. Turns out that by inspecting the entropy term (since the energy is the same for all of them), we can also relate those lower bounds to one another for a given variational distribution $q$, resulting in $\left.\begin{array}{r} F_\lambda^{\text{MAP}}(q) \\ F_\lambda^{\text{marginal}^{\text{U}}}(q) \end{array}\right\} \leq F_\lambda^{\text{MMAP}}(q) \leq F_\lambda^{\textbf{planning}}(q) \leq F_\lambda^{\text{marginal}}(q).$

This in turn means that for the optimal variational distribution of each type of inference we have

$$\left.\begin{array}{r} F_\lambda^{\text{MAP}} \\ F_\lambda^{\text{marginal}^{\text{U}}} \end{array}\right\} \leq F_\lambda^{\text{MMAP}} \leq F_\lambda^{\textbf{planning}} \leq F_\lambda^{\text{marginal}}. \tag{8}$$

See Appendix G for proof. VI aims to maximize a lower bound on the quantity of interest, with tighter bounds generally indicating better performance. Since MMAP is, among the lower bounds, the tightest, it follows that MMAP inference is expected to be no worse and potentially better than all other common types of inference. However, as noted in Section 2.2, MMAP inference is particularly hard, even in trees, meaning that in the case of a non-factorial MDP like the one in Fig. 1[Left], the computation of $F_\lambda^{\text{MMAP}}$ is intractable, even though all the other quantities, including the one of interest, $F_\lambda^{\text{planning}}$, are exactly computable. What we can tractably compute is the lower bound $F_\lambda^{\text{MMAP}}(q) \leq F_\lambda^{\text{MMAP}}$ and try to maximize it wrt $q$, but without guarantees of finding the optimal value. Thus, among the common inference types, MMAP seems a better choice, but it is either intractable or, if using VI, can run into local minima problems. This seems more acceptable in the factored MDP case, but it is disappointing that the problem persists for standard, non-factored MDPs.

### 4.2 The stochasticity of the dynamics is key

The energy term Eq. (3), which is common to all inference methods, contains subterms $\langle \log P(x_{t+1}|a_t, x_t) \rangle_{q(x_{t+1}, x_t, a_t)}$ (and $\langle \log P(x_1) \rangle_{q(x_1)}$ for the first state). When dynamics are deterministic (which we assume to also imply that $P(x_1)$ is deterministic, i.e., the first state is known), this forces the optimal variational conditional to be $q(x_{t+1}|a_t, x_t) = P(x_{t+1}|a_t, x_t)$ (and $q(x_1) = P(x_1)$ for the first state), since any other choice would make those subterms, and therefore the bound, $-\infty$. This affects the relationships of the quantities of interest, which are now (proof in Appendix G):

$$F_\lambda^{\text{marginal}^{\text{U}}} \leq F_\lambda^{\text{MAP}} = F_\lambda^{\text{MMAP}} = F_\lambda^{\textbf{planning}} \leq F_\lambda^{\text{marginal}},$$

and *justifies the use of MAP and MMAP inference as planning when dynamics are deterministic*. When using approximate inference, if dynamics are close to deterministic, it might make more sense to choose the type of inference based on the quality of the approximation, rather than its tightness. If dynamics are stochastic, the suboptimality of MMAP can be explained as a *lack of reactivity to the environment*. Indeed, if we reduce the planning problem to a non-reactive policy $\pi(a_t|x_t) = \pi(a_t)$ we recover MMAP inference as optimal. We test this experimentally in Section 6 and further expand on it in Appendix I.2. MAP has the same problem, but additionally lacks integration over observation sequences ("trajectories"). Even with deterministic dynamics, marginal inference might not produce good utility estimates, but its action posterior will be proportional to the reward of the action sequence, so if we additionally assume $\exp(\lambda R(x)) \in \{0, 1\}$ (i.e., pure planning where we want to attain any of a subset of states), it will produce optimal planning *choices*. Interestingly, our framework also shows that marginal inference is exact for a generalization of MaxEnt planning when the policy entropy regularization is set to $\alpha = 1/\lambda$, regardless of stochasticity, see Appendix E.

# 5 Related work

As stated, the meaning of "planning as inference" is uneven across the literature. (Toussaint and Storkey, 2006) introduce the policy in the MDP factor graph and maximize the likelihood wrt to its parameters using EM. This is an exact approach, although it is more appropriate to say that it is planning as *learning* rather than a type of inference, since the EM process updates the parameters of the factor graph and inference typically operates on a graph with fixed parameters. (Levine, 2018) is a well-known reference that considers *MAP inference* for standard planning and *marginal inference* for MaxEnt planning (Ziebart, 2010). Both are exact only under deterministic dynamics. This problem is not addressed in the case of standard planning, but it is pursued for MaxEnt planning. To achieve exact MaxEnt planning under stochastic dynamics, a modified marginal inference procedure is provided. It can be seen as structured variational inference where $q(x_{t+1}|x_t, a_t) = P(x_{t+1}|x_t, a_t)$ is forced. With the right smoothing, our VBP corresponds to MaxEnt planning, and extends this modified marginal inference to the factored case, see Appendix E.3. (Cui et al., 2015) introduces ARollout, which can be seen as running a single-forward-pass LBP to approximate *marginal inference* for each possible initial action, and then choosing the highest scoring initial action[5], and is applicable to factored MDPs. In the follow-up works (Cui and Khardon, 2016; Cui et al., 2019) the authors develop conformant SOGBOFA, which approximates *marginal-MAP* inference by using ARollout in an inner loop and gradient descent to optimize over the action prior in an outer loop. A number of refinements are added for superior performance. This is a strong baseline and was the runner-up in the international probabilistic planning competition (IPPC) 2018, which agrees with our analysis from Section 4. (Lee et al., 2014; Lee et al., 2016) provide initial results on the connection between conformant planning and MMAP inference. Many works, such as (Attias, 2003) choose *MAP inference* for planning.

Two frameworks (Palmieri et al., 2022; Wu and Khardon, 2022) have been recently introduced to analyze planning from a message-passing perspective. The former analyzes six update rules and their qualitative effect on the plans; the latter focuses on disentangling the inference direction[6] (either forward —from causes to outcomes— or backward —from outcomes to causes—) from the *approximation* type. This work provides two novel message-passing algorithms for factored MDPs: MFVI (mean field VI) and CSVI (collapsed state VI), using the planning tasks from IPPC 2011 as benchmark. We will compare with their results in Section 6.

*Influence diagrams* (Matheson, 2005; Shachter, 2007) are used to represent general decision problems, and various approximate inference approaches have been developed, e.g., (Lee et al., 2018; Lee

---

[5]If the choice of initial action is included in the inference process (rather than in an outer loop), it becomes an MMAP problem. However, this is a "degenerate" MMAP problem with a single maximization variable. To show this degeneracy, consider a non-factored MDP. Exact ARollout is tractable in the non-factored case. In contrast, exact SOGBOFA is not tractable even in the non-factored case because it maximizes over all decision variables.

[6]Note that the term *inference direction* may be misleading: the authors establish a direct equivalence between forward and MMAP inference, and between backward and marginal inference, regardless of the direction in which messages are passed or any other considerations (R. Khardon, personal communication, Oct 2024).

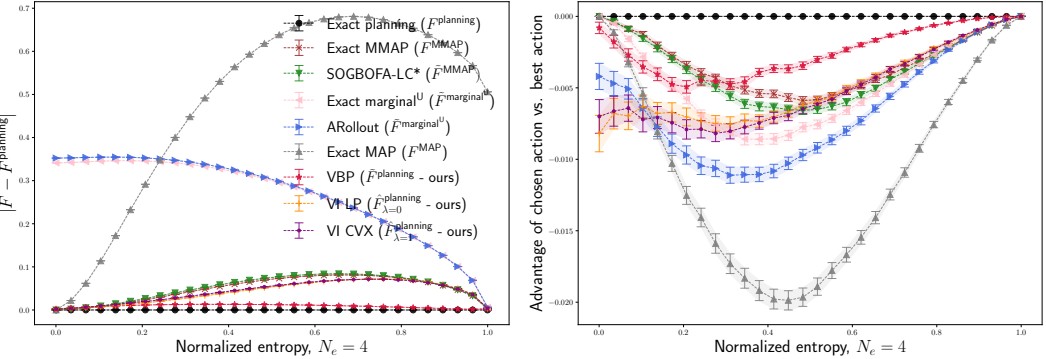

Figure 2: Performance of different types of inference on factored MDPs as a function of their level of stochasticity (normalized entropy). [Left] Estimation error of the best utility. Lower is better. [Right] Advantage of the next action prescribed by a method vs. optimal planning. Higher is better.

et al., 2020). Closest to our work, (Cheng et al., 2013; Chen et al., 2015) tackle graph-based MDPs, similar to factored MDPs, but with a factorized action space: multiple actions are taken at each time step, each locally affecting a single entity. This locality results in additional efficiencies, so direct application to a state-only-factored MDPs would still result in exponential cost.

LP formulations for the solution non-stationary, finite-horizon MDPs have received significant attention over the last decade (e.g., Kumar et al., 2015; Bhattacharya and Kharoufeh, 2017; Altman, 2021; Bhat et al., 2023), but they lack a variational perspective and do not generalize easily to handle state-factored MDPs. The LPs in (Koller and Parr, 1999; Guestrin et al., 2003; Malek et al., 2014) on the other hand do handle factored MDPs and have a closer connection to our work. The problem setup is slightly different, infinite-horizon MDPs with a stationary policy in their case vs our finite horizon MDPs, which allows us to use *local* non-stationary policies. More importantly, their computational cost can be significantly higher than in this proposal. E.g., (Guestrin et al., 2003, Section 4.2.1) states that the cost is dependent on the variable elimination order. In the optimal case (which is NP-hard to find), it scales exponentially with the *width of the cost network*, which is based on the dependencies between entities, and can be much larger than exponential in the number of parents (our case).

# 6 Empirical validation

**Synthetic MDPs**  We generate $5,000$ synthetic factored MDPs structured as in Fig. 1[Right] with random dynamics, all-or-nothing reward at the last time step, and controlled normalized entropies, defined as $H_{\mathrm{MDP}} = \frac{\sum_{i,x_t,a_t} H(p(x_{t+1}^{(i)}|x_t,a_t))}{N_e N_a N_s^{N_e} \log N_s} \in [0,1]$. See Appendix I.1 for more details.

They are purposefully small so that we can compute $F^{\mathrm{marginal^U}}$, $F^{\mathrm{MAP}}$, $F^{\mathrm{MMAP}}$, $F^{\mathrm{planning}}$ exactly, even though they are intractable in general. We also compute $\tilde{F}^{\mathrm{marginal}}$ (tractable), and $\tilde{F}^{\mathrm{MMAP}}$ (tractable bound, generally intractable optimization), which correspond to ARollout (Cui et al., 2015) and optimal SOGBOFA-LC* (Cui et al., 2019), respectively. Finally, we include VBP (tractable, imperfect optimization of $\tilde{F}^{\mathrm{planning}}(\boldsymbol{q})$) and the tractable VI LP $\hat{F}^{\mathrm{planning}}_{\lambda=0}$ and VI CVX $\hat{F}^{\mathrm{planning}}_{\lambda=1}$.

Fig. 2 shows the effect of stochasticity in the estimation of the utility [Left] and the next best action [Right]. For high stochasticity, VBP, even if approximate, dominates all other types of inference. The concave upper bounds $\hat{F}^{\mathrm{planning}}_{\lambda=0}$ and $\hat{F}^{\mathrm{planning}}_{\lambda=1}$ also improve over exact MMAP but not as much as VBP. For low stochasticity, exact MAP and MMAP dominate VBP. The (intractably optimal) SOGBOFA-LC* remains close to the exact MMAP. These results agree well with the theory and observations laid out in Section 4. We see good correlation between the accuracy of the utility estimation and the quality of the planning choices ([Left] and [Right] panels). ARollout and exact marginal seem to be an exception to this; this is explained in Section 4.2: for pure planning problems, with low stochasticity both methods are a constant away from the right utility and make good choices.

**Reactivity avoidance**  We craft a multi-entity MDP in which the agent controls the level of reactivity (see Section 4.2) needed to solve the environment, but is penalized for lower ones. VBP keeps the reactivity at a maximum, to achieve a reward of 1. SOGBOFA-LC* "aware" that it cannot plan reactively (despite replanning), takes step to reduce it, getting a reward of 0.33. See Appendix I.2.

**International probabilistic planning competition tasks (IPPC)**  We follow (Wu and Khardon, 2022) and compare on the same tasks and with the same methods. We use the 6 different domains from IPPC2011, each with 10 instances (factored MDPs) of increasing difficulty, with given dynamics and (stationary) rewards, 40-step episodes, and mildly stochastic dynamics. As baselines, we use MFVI-Bwd (Wu and Khardon, 2022), CSVI-Bwd (Wu and Khardon, 2022), ARollout ($\tilde{F}^{\mathrm{marginal^U}}_{\lambda=0}$, see Cui et al., 2015), and SOGBOFA-LC ($\tilde{F}^{\mathrm{MMAP}}_{\lambda=0}$, see Cui et al., 2019). We provide details about these competing methods in Appendix I.3. From our proposed variational framework, we use[7] VI LP ($\hat{F}^{\mathrm{planning}}_{\lambda=0}$), and VBP[8]. ($\tilde{F}^{\mathrm{planning}}_{\lambda\approx0}$).

Fig. 3 shows the average cumulative reward for all domains and methods. Four domains are highly deterministic ($H_{\mathrm{MDP}} < 0.05$), but planning inference manages to be competitive wrt the best baselines.

---

[7]Code at `https://github.com/google-deepmind/what_type_of_inference_is_planning`.

[8]Setting $\lambda = 0$ can result in degeneracy problems, so we use a small value instead, see Appendix I.3.

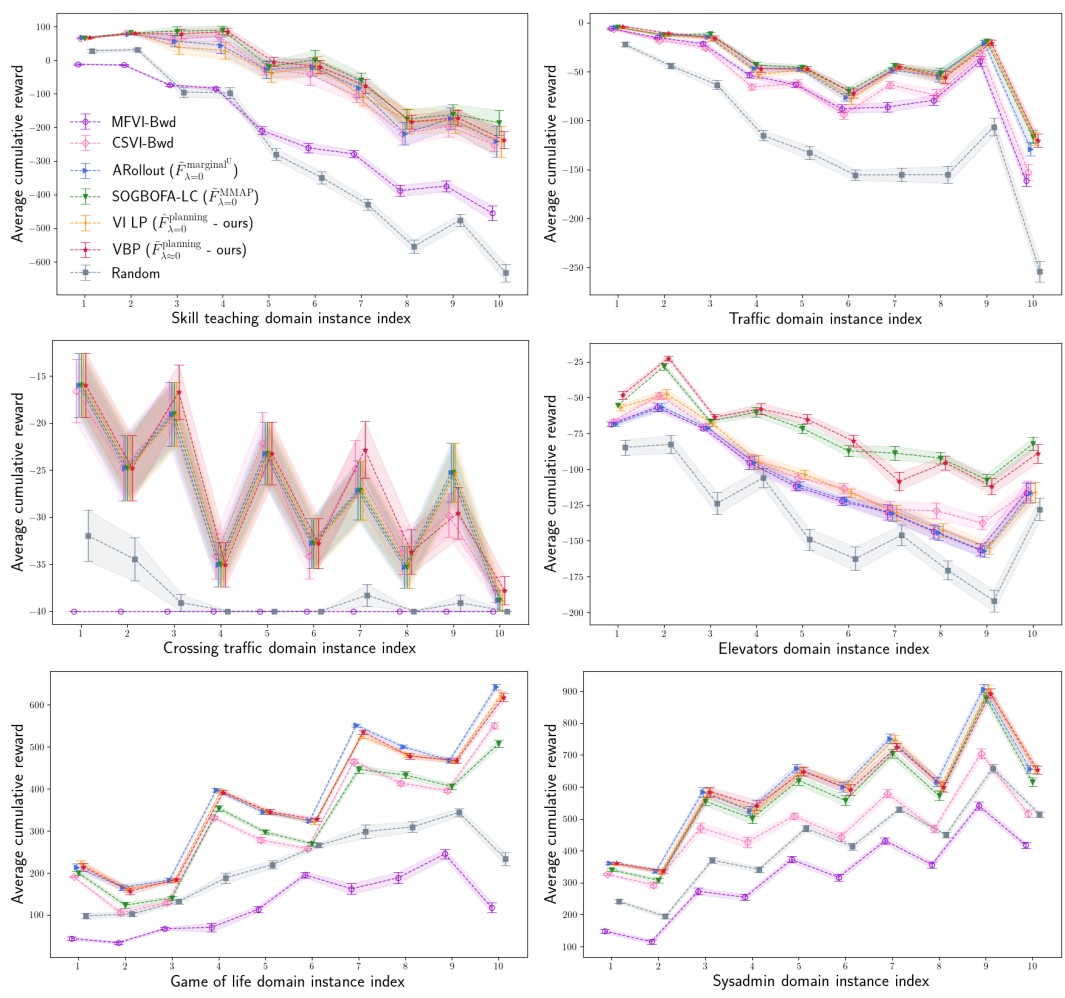

Figure 3: Cumulative rewards on 6 problem domains from the ICAPS 2011 IPPC. A small horizontal jitter was introduced in all data points for visual clarity. Each cumulative reward is averaged over 30 simulations per instance. Datasets are ordered from left to right and top to bottom by increasing normalized entropy levels. Only the last two have a significant stochasticity level >5%.

The other two are Game of Life and SysAdmin, which have an average $H_{\mathrm{MDP}}$ of 0.18 and 0.23 respectively. We notice a significant advantage of our proposals wrt the most sophisticated method, SOGBOFA-LC ($\tilde{F}^{\mathrm{MMAP}}_{\lambda=0}$). This is consistent with our expectation of MMAP degrading with increased stochasticity (see Section 4). Elevators is well known for its challenging rewards (Cui et al., 2015) and the only one for which ARollout performs noticeably worse. In this domain, VBP manages to match or exceed SOGBOFA-LC on most instances. Overall, we observe that VBP is more consistent across varying stochasticities, matching the performance of the best method for each dataset. None of these domains reach the larger stochasticity levels shown in Fig. 2 where VBP dominates. VI LP also performs generally well, although not as well as VBP due to the missing mutual information mentioned in Section 3.2. See Appendix I.3 for details.

# 7 Discussion

The variational framework offers a powerful tool to analyze and understand how different existing types of inference approximate planning, the key role of stochasticity, what the ideal type of inference for planning is, and how to design new approximations. We hope that the introduced VI perspective will further the understanding of existing methods and lead to novel planning algorithms.

## Acknowledgements

We thank Roni Khardon, Junkyu Lee, and the anonymous referees for their feedback on an earlier version of this paper, which helped us to improve its clarity and presentation.

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

# A    Proof of the variational formulation of planning

In this proof we will use two identities, the first is the variational identity (Jordan et al., 1999):

$$\log \sum_{\boldsymbol{x},\boldsymbol{a}} f(\boldsymbol{x},\boldsymbol{a}) = \max_{q(\boldsymbol{x},\boldsymbol{a})} \langle \log f(\boldsymbol{x},\boldsymbol{a}) \rangle_{q(\boldsymbol{x},\boldsymbol{a})} + H(q(\boldsymbol{x},\boldsymbol{a})) \tag{9}$$

and the second is

$$\max_{\pi} \sum_{\boldsymbol{a}} q(\boldsymbol{a}) \log \pi(\boldsymbol{a}) = \max_{\pi(\boldsymbol{a})} -H(q(\boldsymbol{a})) - \mathrm{KL}(q(\boldsymbol{a})||\pi(\boldsymbol{a})) = \sum_{\boldsymbol{a}} q(\boldsymbol{a}) \log q(\boldsymbol{a}), \tag{10}$$

which follows because the term $\pi(\boldsymbol{a})$ is an arbitrary distribution that can make the KL divergence exactly zero (by choosing $\pi(\boldsymbol{a}) = q(\boldsymbol{a})$). With this we can proceed to the main proof:

$$F_\lambda^{\mathrm{planning}} = \frac{1}{\lambda} \max_{\boldsymbol{\pi}} \log \sum_{\boldsymbol{a},\boldsymbol{x}} \exp\left(\lambda \sum_{t=1}^{T-1} R_t(x_t,a_t,x_{t+1})\right) P(x_1) \prod_{t=1}^{T-1} P(x_{t+1}|a_t,x_t)\pi_t(a_t|x_t)$$

$$\overset{\mathrm{Eq.\ (9)}}{=} \frac{1}{\lambda} \max_{\boldsymbol{\pi},\boldsymbol{q}} \left\langle \log\left(P(x_1) \prod_{t=1}^{T-1} \exp(\lambda R_t(x_t,a_t,x_{t+1})) P(x_{t+1}|a_t,x_t)\pi_t(a_t|x_t)\right)\right\rangle_{q(\boldsymbol{x},\boldsymbol{a})} + H(q(\boldsymbol{x},\boldsymbol{a}))$$

$$= \frac{1}{\lambda} \max_{\boldsymbol{\pi},\boldsymbol{q}} \left(-E_\lambda(\boldsymbol{q}) + H(q(\boldsymbol{x},\boldsymbol{a})) + \sum_{t=1}^{T-1} \langle \log \pi_t(a_t|x_t) \rangle_{q(x_t,a_t)}\right)$$

$$= \frac{1}{\lambda} \max_{\boldsymbol{q}} \left(-E_\lambda(\boldsymbol{q}) + H_q(x_1) + \sum_{t=1}^{T-1} H_q(x_{t+1},a_t|x_t) + \langle \max_{\pi_t} \langle \log \pi_t(a_t|x_t) \rangle_{q(a_t|x_t)} \rangle_{q(x_t)}\right)$$

$$\overset{\mathrm{Eq.\ (10)}}{=} \frac{1}{\lambda} \max_{\boldsymbol{q}} \left(-E_\lambda(\boldsymbol{q}) + H_q(x_1) + \sum_{t=1}^{T-1} H_q(x_{t+1},a_t|x_t) + \langle\langle \log q(a_t|x_t) \rangle_{q(a_t|x_t)} \rangle_{q(x_t)}\right)$$

$$= \frac{1}{\lambda} \max_{\boldsymbol{q}} \left(-E_\lambda(\boldsymbol{q}) + H_q(x_1) + \sum_{t=1}^{T-1} H_q(x_{t+1},a_t|x_t) - H_q(a_t|x_t)\right)$$

$$= \frac{1}{\lambda} \max_{\boldsymbol{q}} \left(-E_\lambda(\boldsymbol{q}) + H^{\mathrm{planning}}(\boldsymbol{q})\right) = \max_{\boldsymbol{q}} F_\lambda^{\mathrm{planning}}(\boldsymbol{q}),$$

where we additionally see that we have chosen $\pi_t(a_t|x_t) = q(a_t|x_t)$, therefore, the optimal policy will be $\pi_t(a_t|x_t) = q^*(a_t|x_t)$ where $q^*$ is the value of $q$ that maximizes $F_\lambda^{\mathrm{planning}}(\boldsymbol{q})$.

# B    Proof that for $\lambda \to 0^+$ the variational bound turns into an LP

Let us rewrite the concave optimization functional Eq. (2) (derived in Appendix A) by combining the energy and entropy terms into KL divergences:

$$F_\lambda^{\mathrm{planning}}(\boldsymbol{q}) = \frac{1}{\lambda}\left(-E_\lambda(\boldsymbol{q}) + H^{\mathrm{planning}}(\boldsymbol{q})\right) = \frac{1}{\lambda}\left(-\mathrm{KL}(q(x_1)||P(x_1))\right.$$

$$\left. - \sum_{t=1}^{T-1} \langle \mathrm{KL}(q(x_{t+1}|x_t,a_t)||P(x_{t+1}|x_t,a_t)) \rangle_{q(x_t,a_t)} + \langle \lambda R_t(x_t,a_t,x_{t+1}) \rangle_{q(x_{t+1},x_t,a_t)}\right)$$

$$= \sum_{t=1}^{T-1} \langle R_t(x_t,a_t,x_{t+1}) \rangle_{q(x_{t+1},x_t,a_t)}$$

$$- \frac{\mathrm{KL}(q(x_1)||P(x_1)) + \sum_{t=1}^{T-1} \langle \mathrm{KL}(q(x_{t+1}|x_t,a_t)||P(x_{t+1}|x_t,a_t)) \rangle_{q(x_t,a_t)}}{\lambda}$$

It is clear that if any of the KL terms are larger than 0, then $\lim_{\lambda\to 0^+} F_\lambda^{\mathrm{planning}}(\boldsymbol{q}) = -\infty$ (for bounded rewards), whereas if all KL terms are 0, the limit is finite. That means that to maximize the bound wrt $\boldsymbol{q}$ in the $\lambda \to 0^+$ limit, we will choose $q(x_1) = P(x_1)$ and $q(x_{t+1}|x_t,a_t) = P(x_{t+1}|x_t,a_t)$, which allows to remove the KL terms and results in the (constrained) LP optimization of Corollary 1.1 (which also explicitly includes the marginalization constraints $\boldsymbol{q} \in \mathcal{M}$).

## C Derivation of the planning Bethe entropy for factored MDPs

The standard Bethe entropy of a factored MDP (Yedidia et al., 2005), such as the one in Fig. 4, is:

$$H_{\text{Bethe}}^{\text{marginal}}(\tilde{\boldsymbol{q}}) = \sum_{i=1}^{N_e} H_q(x_1^{(i)}) + \sum_{t=1}^{T-1} \left( H_{\text{Bethe}}(\tilde{\boldsymbol{q}}_{x_t, a_t}) - \sum_{i=1}^{N_e} H_q(x_t^{(i)}) \right.$$
$$\left. + \sum_{i=1}^{N_e} H_q(x_{t+1}^{(i)}, x_t^{\text{pa}(i)}, a_t) - H_q(x_t^{\text{pa}(i)}, a_t) \right),$$

where we use $x_t^{\text{pa}(i)}$ to refer to the "parent" variables. To simplify notation, when $i \in 1, \ldots, N_e$, $x_t^{\text{pa}(i)}$ is the collection variables on which the distribution of $x_{t+1}^{(i)}$ depends according to the dynamics model, but when $i \in N_e + 1, \ldots, N_e + N_r$, $x_t^{\text{pa}(i)}$ is the collection of variables on which the $(i - N_e)$-th reward depends. See also Fig. 4 for clarification on the notation of parents of dynamics and rewards.

The Bethe entropy above was defined, for conciseness, in terms of the Bethe entropy of a subset of the variables in a single time slice (current state and action, but not next state):

$$H_{\text{Bethe}}(\tilde{\boldsymbol{q}}_{x_t, a_t}) = (1 - N_e) H_q(a_t) + \sum_{i=1}^{N_e} H_q(x_t^{\text{pa}(i)}, a_t) + \sum_{i=N_e+1}^{N_e+N_r} H_q(x_t^{\text{pa}(i)})$$
$$- \sum_{i=1}^{N_e+N_r} \sum_{k \in \text{pa}(i)} H_q(x_t^{(k)}) + \sum_{i=1}^{N_e} H_q(x_t^{(i)}).$$

Finally, the Bethe entropy of the states of factored MDP is

$$H_{\text{Bethe}}(\tilde{\boldsymbol{q}}_{x_t}) = \sum_{i=1}^{N_e} H_q(x_t^{(i)}) + \sum_{i=1}^{N_e+N_r} \left( H_q(x_t^{\text{pa}(i)}) - \sum_{k \in \text{pa}(i)} H_q(x_t^{(k)}) \right)$$
$$= \sum_{i=1}^{N_e} H_q(x_t^{(i)}) - \sum_{i=1}^{N_e+N_r} I_q(x_t^{\text{pa}(i)})$$

where $I_q(x_t^{\text{pa}(i)})$ is the mutual information among the parents of variable $x_{t+1}^{(i)}$.

Note that all the Bethe entropy definitions are linear combinations of standard Shannon entropies defined over subsets of variables in the factor graph. The subsets are defined by the factor graph, as groups of variables connected by the same factor. The idea of the Bethe entropy approximation is to sum the entropies of all such subsets and then discount the "overcounted" entropy corresponding to variables that appear in multiple subsets. The pseudo-marginals $\tilde{\boldsymbol{q}} \equiv \{q(x_{t+1}^{(i)}, x_t^{\text{pa}(i)}, a_t)\}_{t=1, i=1}^{t=T-1, i=N_e} \cup \{q(x_t^{\text{pa}(r)})\}_{t=1, r=N_e+1}^{t=T, r=N_e+N_r}$ are all the local distributions that correspond to each factor. The pseudo-marginals are locally consistent at the variables, i.e., two pseudo-marginals that contain the same variable should provide the same marginal for that variable. However, there is no further need for consistency, and in particular, they do not need to correspond to the marginals of global joint distribution. The (convex) domain of pseudo-marginals contains the (also convex) domain of the marginals, but it is larger, so switching from marginals to pseudo-marginals in any optimization problem relaxes it and provides an upper bound on the original optimization problem. See (Weller et al., 2014) for more details. For convenience, we have also defined some subsets of the pseudo-marginals: $\tilde{\boldsymbol{q}}_{x_t, a_t} \equiv \{q(x_t^{\text{pa}(i)}, a_t)\}_{i=1}^{i=N_e} \cup \{q(x_t^{\text{pa}(r)})\}_{r=N_e+1}^{r=N_e+N_r}$ and $\tilde{\boldsymbol{q}}_{x_t} \equiv \{q(x_t^{\text{pa}(i)})\}_{i=1}^{i=N_e+N_r}$.

Since the planning entropy is a linear combination of the standard entropy of the factor graph $H^{\text{marginal}}(\boldsymbol{q})$ and two local entropies per time step, we can simply approximate each Shannon entropy

by its corresponding Bethe entropy to get the Bethe planning entropy

$$H^{\text{planning}}(\boldsymbol{q}) = H^{\text{marginal}}(\boldsymbol{q}) + \sum_{t=1}^{T-1} H_q(x_t) - H_q(x_t, a_t)$$

$$\approx H^{\text{marginal}}_{\text{Bethe}}(\tilde{\boldsymbol{q}}) + \sum_{t=1}^{T-1} H_{\text{Bethe}}(\tilde{\boldsymbol{q}}_{x_t}) - H_{\text{Bethe}}(\tilde{\boldsymbol{q}}_{x_t,a_t})$$

$$= \sum_{i=1}^{N_e} H_q(x_1^{(i)}) + \sum_{t=1}^{T-1} \Big( \sum_{i=1}^{N_e} H_q(x_{t+1}^{(i)}|x_t^{\text{pa}(i)}, a_t) - \sum_{i=1}^{N_e+N_r} I_q(x_t^{\text{pa}(i)}) \Big) = H^{\text{planning}}_{\text{Bethe}}(\tilde{\boldsymbol{q}}).$$

Note that we sometimes use the superscript marginal for consistency with the main text, but the marginal entropy is simply the standard entropy so that superscript can be safely dropped.

## D    Value BP message updates

We are interested in optimizing the cost function

$$\frac{1}{\lambda} \max_{\tilde{\boldsymbol{q}}} \Big( -E_\lambda(\tilde{\boldsymbol{q}}) + \epsilon H^{\text{marginal}}_{\text{Bethe}}(\tilde{\boldsymbol{q}}) + (1-\epsilon) H^{\text{planning}}_{\text{Bethe}}(\tilde{\boldsymbol{q}}) \Big)$$

As we saw in Appendix C, $H^{\text{marginal}}_{\text{Bethe}}(\tilde{\boldsymbol{q}})$ and $H^{\text{planning}}_{\text{Bethe}}(\tilde{\boldsymbol{q}})$ are linear combinations of local entropies over small subsets of variables defined by the factor graph, therefore so is their linear combination. This score function can be seen as a standard Bethe free energy with non-standard entropy weightings. We can directly derive LBP-like message updates by using these modified weights instead (Hazan and Shashua, 2010), resulting in the following VBP message updates

$$Q(x_t^{\text{pa}(i)}, a_t) = \sum_{x_{t+1}^{(i)}} m_{\text{b}}(x_{t+1}^{(i)}) p(x_{t+1}^{(i)}|x_t^{\text{pa}(i)}, a_t)$$

$$m_{\text{b}}(x_t^{\text{pa}(i)}) = \Big( \sum_a (Q(x_t^{\text{pa}(i)}, a_t) n^{(i)}(a_t))^{\frac{1}{\epsilon}} \Big)^\epsilon$$

$$m^{(i)}(a_t) = \Big( \sum_{x_t^{\text{pa}(i)}} \Big( \frac{Q(x_t^{\text{pa}(i)}, a_t)}{m_{\text{b}}(x_t^{\text{pa}(i)})} \Big)^{\frac{1}{\epsilon}} m_{\text{f}}(x_t^{\text{pa}(i)}) m_{\text{b}}(x_t^{\text{pa}(i)}) \Big)^\epsilon$$

$$n^{(i)}(a_t) = \prod_{k \neq i} m^{(k)}(a_t)$$

$$m_{\text{f}}(x_{t+1}^{(i)}) = \sum_{x_t^{\text{pa}(i)}, a} \Big( \frac{Q(x_t^{\text{pa}(i)}, a_t) n^{(i)}(a_t)}{m_{\text{b}}(x_t^{\text{pa}(i)})} \Big)^{\frac{1}{\epsilon}} m_{\text{f}}(x_t^{\text{pa}(i)}) m_{\text{b}}(x_t^{\text{pa}(i)}) \frac{p(x_{t+1}^{(i)}|x_t^{\text{pa}(i)}, a_t)}{Q(x_t^{\text{pa}(i)}, a_t)}$$

$$m_{\text{f}}(x_t^{\text{pa}(i)}) = \prod_{k \in \text{pa}(i)} n_{\text{f}}^{(i)}(x_t^{(k)})$$

$$n_{\text{f}}^{(i)}(x_t^{(j)}) = m_{\text{f}}(x_t^{(j)}) \prod_{k|j \in \text{pa}(k), k \neq i} n_{\text{b}}^{(k)}(x_t^{(j)}) \quad \text{towards parents of entity } i$$

$$n_{\text{b}}^{(i)}(x_t^{(j)}) = \sum_{\{x_t^{(k)}\}_{k \neq j}} m_{\text{b}}(x_t^{\text{pa}(i)}) \prod_{k \neq i} n_{\text{f}}^{(i)}(x_t^{(k)}) \quad \text{from parents of entity } i$$

$$m_{\text{b}}(x_t^{(j)}) = \prod_{k|j \in \text{pa}(k)} n_{\text{b}}^{(k)}(x_t^{(j)})$$

The following messages should be held constant

$$m_{\text{b}}(x_t^{\text{pa}(i)}) = \exp(\lambda R(x_t^{\text{pa}(i)})) \ \forall i > N_e$$

$$m_{\text{f}}(x_1^{(i)}) = P(x_1^{(i)})$$

See Fig. 4 to track the correspondence between the above messages and the factor graph of the factored MDP. Note that most updates correspond exactly with standard loopy BP, and only a few (the ones involving $\epsilon$) are specific to VBP. These messages updates should be iterated until convergence or for a fixed number of iterations. Two tricks to improve convergence are (a) damping the message updates in log space; (b) as the iterations progress, anneal between LBP ($\epsilon = 1$) and VBP (very small $\epsilon$). We typically use both, with a damping of 0.5 (i.e., the mean of the old and the new message in log space) and anneal by using as $\epsilon(\text{iter}) = \max\{0.01, 1.0/\text{iter}\}$, where "iter" is the iteration number. Scheduling also plays a role in convergence. We propagate the messages by alternating backward and forward schedules in an outer loop, and solving each time slice to convergence in an inner loop. We did observe a correlation between the quality of the solutions and VBP converging.

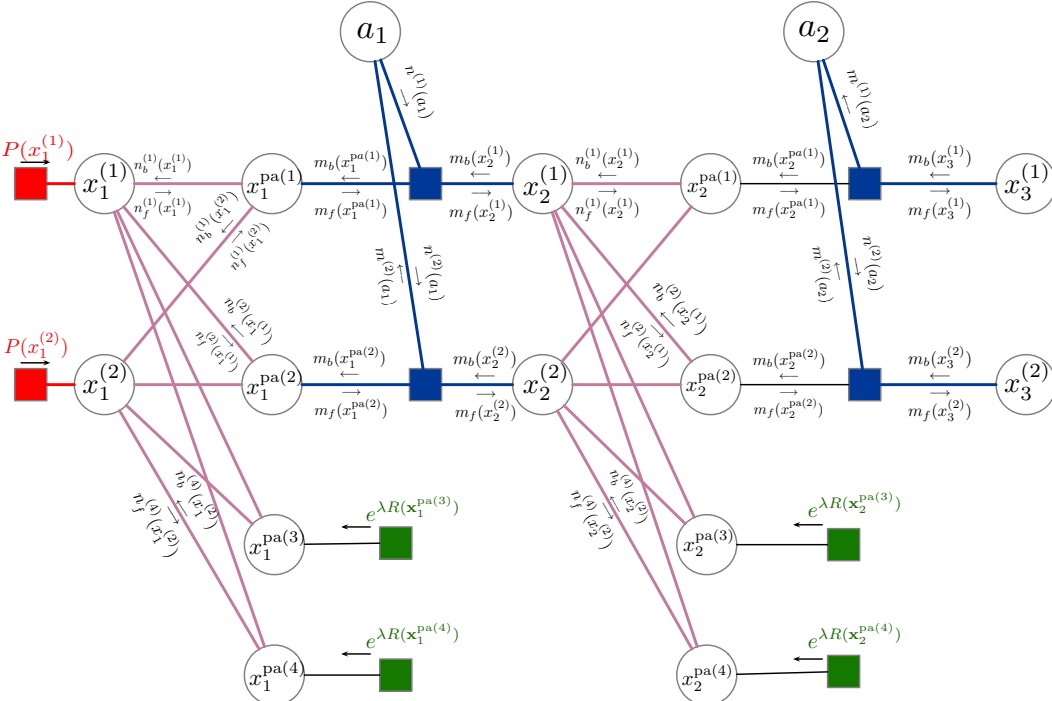

Figure 4: Correspondence between the message passing updates and the factorized MDP.

## E   The connection with maximum entropy reinforcement learning

If we assume, as we have done throughout this paper, that the reward and dynamics functions are known, maximum entropy reinforcement learning (MERL) corresponds to finding the policy that maximizes a weighted combination of the reward and the policy entropy, averaged over the trajectories induced by such policy. In this restricted setting, some of the main difficulties of RL disappear (e.g., how to efficiently explore and discover the reward and dynamics functions), and "MaxEnt planning" might be a more precise term. However, we stick in this section to the more common term MERL as used in the literature (e.g., Levine, 2018), even when only discussing planning. MERL maximizes the following objective:

$$\max_{\boldsymbol{\pi}}\langle R(\boldsymbol{x},\boldsymbol{a}) + \alpha H(\pi(\boldsymbol{a}|\boldsymbol{x}))\rangle_{P(\boldsymbol{x}|\boldsymbol{a})\pi(\boldsymbol{a}|\boldsymbol{x})} = \max_{\boldsymbol{\pi}}\langle R(\boldsymbol{x},\boldsymbol{a}) - \alpha \log \pi(\boldsymbol{a}|\boldsymbol{x})\rangle_{P(\boldsymbol{x}|\boldsymbol{a})\pi(\boldsymbol{a}|\boldsymbol{x})},$$

where $\alpha$ controls the policy regularization level. For $\alpha = 0$ we recover standard planning, and as $\alpha \to \infty$, the optimal policy tends to the uniform policy.

We can define a $\lambda$-generalized version of MERL:

$$F_\lambda^{\text{MERL}} = \frac{1}{\lambda}\max_{\boldsymbol{\pi}}\log\langle\exp(\lambda[R(\boldsymbol{x},\boldsymbol{a}) - \alpha \log \pi(\boldsymbol{a}|\boldsymbol{x})])\rangle_{P(\boldsymbol{x}|\boldsymbol{a})\pi(\boldsymbol{a}|\boldsymbol{x})}.$$

With this definition, when $\lambda \to 0^+$, we recover the standard MERL objective, and when $\alpha \to 0^+$, we recover Eq. (1), standard planning with an exponential utility function parameterized by $\lambda$.

Following the same steps as in Appendix A, but including the "policy regularization" term $-\alpha \log \pi(\boldsymbol{a}|\boldsymbol{x})$, we can find the corresponding variational form of the $\lambda$-generalized MERL:

$$F_\lambda^{\mathrm{MERL}} = \frac{1}{\lambda} \max_{\boldsymbol{\pi}} \log \langle \exp(\lambda[R(\boldsymbol{x}, \boldsymbol{a}) - \alpha \log \pi(\boldsymbol{a}|\boldsymbol{x})]) \rangle_{P(\boldsymbol{x}|\boldsymbol{a})\pi(\boldsymbol{a}|\boldsymbol{x})}$$

$$= \frac{1}{\lambda} \max_{\boldsymbol{q}} \Big( - E_\lambda(\boldsymbol{q}) + (1-\epsilon)H^{\mathrm{planning}}(\boldsymbol{q}) + \epsilon H^{\mathrm{marginal}}(\boldsymbol{q}) \Big) = \max_{\boldsymbol{q}} F_\lambda^{\mathrm{MERL}}(\boldsymbol{q})$$

where we use the shorthand $\epsilon = \alpha\lambda$ and need to assume $\epsilon \leq 1$ for the above equality to hold (or equivalently, $\alpha \leq 1/\lambda$). This is an interesting result that shows that policy regularization corresponds to a variational objective that interpolates between planning and marginalization, with both entropies being precisely defined in Table 1. This in turn means that, when the $\lambda$-generalized MERL uses a finite $\lambda > 0$, *there is a policy regularization level $\alpha = 1/\lambda$ for which the variational posterior for $\lambda$-generalized MERL coincides exactly with marginal inference.*

One can view the smoothed value BP message-passing updates from Appendix D from two alternative but perfectly equivalent perspectives: In one, we smooth the planning entropy (which can lead to ill-defined messages) with the standard marginal entropy from belief propagation. In the other, the smoothing comes from regularizing the reward with the policy, i.e., from performing generalized MaxEnt planning instead of standard planning. This is still true when $\lambda \to 0^+$, the case typically considered in the literature when talking about MERL (Ziebart, 2010; Levine, 2018).

### E.1 The connection with Sergey Levine's "RL as probabilistic inference" (the $\lambda \to 0^+$ case)

In the common case in which rewards are additive, we have that $\lambda \to 0^+$ and $\alpha$ is unconstrained. To analyze this case, we first rewrite the $\lambda$-generalized MERL variational objective using KL divergences, as we did in Appendix B for the planning variational objective:

$$F_\lambda^{\mathrm{MERL}}(\boldsymbol{q}) = \langle R(\boldsymbol{x}, \boldsymbol{a}) \rangle_{q(\boldsymbol{x}, \boldsymbol{a})} + \alpha H_q(\boldsymbol{a}|\boldsymbol{x})$$
$$- \frac{\mathrm{KL}(q(x_1)||P(x_1)) + \sum_{t=1}^{T-1} \langle \mathrm{KL}(q(x_{t+1}|x_t, a_t)||P(x_{t+1}|x_t, a_t)) \rangle_{q(x_t, a_t)}}{\lambda}.$$

Following the same reasoning as in Appendix B, since the KL terms are non-negative, the only way for $F_\lambda^{\mathrm{MERL}}(\boldsymbol{q})$ to have a finite value as $\lambda \to 0^+$ is to set $q(x_{t+1}|x_t, a_t) = P(x_{t+1}|x_t, a_t) \; \forall t$, which cancels the KL term and removes the dependence on $\lambda$. Thus,

$$F_{\lambda \to 0^+}^{\mathrm{MERL}} = \max_{\boldsymbol{q}} F_{\lambda \to 0^+}^{\mathrm{MERL}}(\boldsymbol{q})$$
$$= \max_{\boldsymbol{q}} \langle R(\boldsymbol{x}, \boldsymbol{a}) \rangle_{q(\boldsymbol{x}, \boldsymbol{a})} + \alpha H_q(\boldsymbol{a}|\boldsymbol{x}) \text{ s.t. } q(x_{t+1}|x_t, a_t) = P(x_{t+1}|x_t, a_t) \; \forall t, \quad (11)$$

which is a policy-regularized version of Corollary 1.1. We implicitly assume $\boldsymbol{q}$ to be constrained to the space of density functions, instead of including all the linear constraints that guarantee this, as we explicitly did in Corollary 1.1. This objective is concave with linear constraints and therefore has a single maximum, the MERL with additive rewards, $F_{\lambda \to 0^+}^{\mathrm{MERL}}$.

Observe that Eq. (11) corresponds exactly with the structured variational inference from (Levine, 2018), including the constrained form of the posterior (compare with Eq. (19) within (Levine, 2018), where they have used $\alpha = 1$).

We can further show that the optimal posterior can also be found by performing marginal inference with a constrained variational posterior on the *graphical model with $\lambda = 1/\alpha$*, which is not immediately intuitive. Indeed, the optimal solution to problem Eq. (11) is maintained if we scale it. Assuming $\alpha > 0$, we can write

$$\max_{\boldsymbol{q}} F_{\lambda \to 0^+}^{\mathrm{MERL}}(\boldsymbol{q}) = \alpha \max_{\boldsymbol{q}} \frac{1}{\alpha} F_{\lambda \to 0^+}^{\mathrm{MERL}}(\boldsymbol{q}) - \langle \log q(\boldsymbol{x}|\boldsymbol{a}) - \log P(\boldsymbol{x}|\boldsymbol{a}) \rangle_{q(\boldsymbol{x}, \boldsymbol{a})}$$
$$\text{s.t. } q(x_{t+1}|x_t, a_t) = P(x_{t+1}|x_t, a_t) \; \forall t$$
$$= \alpha \max_{\boldsymbol{q}} \langle R(\boldsymbol{x}, \boldsymbol{a})/\alpha + \log P(\boldsymbol{x}|\boldsymbol{a}) \rangle_{q(\boldsymbol{x}, \boldsymbol{a})} + H_q(\boldsymbol{x}, \boldsymbol{a})$$
$$\text{s.t. } q(x_{t+1}|x_t, a_t) = P(x_{t+1}|x_t, a_t) \; \forall t$$
$$= \alpha \max_{\boldsymbol{q}} F_{\lambda = 1/\alpha}^{\mathrm{marginal}}(\boldsymbol{q})$$
$$\text{s.t. } q(x_{t+1}|x_t, a_t) = P(x_{t+1}|x_t, a_t) \; \forall t.$$

The first equality is correct because $\log q(\boldsymbol{x}|\boldsymbol{a}) - \log P(\boldsymbol{x}|\boldsymbol{a}) = 0$, under the assumption $q(x_{t+1}|x_t, a_t) = P(x_{t+1}|x_t, a_t) \; \forall t$, and therefore the added term is 0. The last expression corresponds to standard (marginal) variational inference for the graphical model $\exp(R(\boldsymbol{x}, \boldsymbol{a})/\alpha)P(\boldsymbol{x}|\boldsymbol{a})$. In other words, the optimal $\boldsymbol{q}$ can be obtained by two seemingly unconnected routes: either maximizing $F_{\lambda \to 0^+}^{\text{MERL}}$ with no constraints on the form of the posterior, or maximizing $F_{\lambda=1/\alpha}^{\text{marginal}}$ constraining the posterior dynamics to match the prior dynamics. This is what is done in (Levine, 2018), where they implicitly use $\lambda = \alpha = 1$. In the special case of deterministic dynamics, the constraint of posterior dynamics matching prior dynamics is already enforced by marginal inference, so it does not need to be separately enforced. For this reason, in (Levine, 2018) it is said that for fixed dynamics MERL corresponds to vanilla marginal inference.

## E.2 Maximum-entropy value belief propagation (MaxEnt VBP)

We are interested in optimizing the cost function

$$\tilde{F}_\lambda^{\text{MERL}} = \frac{1}{\lambda} \max_{\tilde{\boldsymbol{q}}} \left( -E_\lambda(\tilde{\boldsymbol{q}}) + \alpha\lambda H_{\text{Bethe}}^{\text{marginal}}(\tilde{\boldsymbol{q}}) + (1-\alpha\lambda) H_{\text{Bethe}}^{\text{planning}}(\tilde{\boldsymbol{q}}) \right) \text{ with } \alpha\lambda \leq 1.$$

which is exactly the same cost function as in Appendix D, but with $\epsilon = \alpha\lambda$ so as to explicitly correspond to generalized MERL. The message updates are essentially the same as in Appendix D, but explicitly revealing the influence of $\lambda$ on $\epsilon$ allows us to define a different message (power-) scale to obtain message updates that are well-defined as $\lambda$ becomes closer to zero. Here are the re-scaled messages, and the updates defined in terms of those re-scaled messages:

$$Q(x_t^{\text{pa}(i)}, a_t)^{1/\lambda} = \bar{Q}(x_t^{\text{pa}(i)}, a_t) = \left( \sum_{x_{t+1}^{(i)}} \bar{m}_{\text{b}}(x_{t+1}^{(i)})^\lambda p(x_{t+1}^{(i)}|x_t^{\text{pa}(i)}, a_t) \right)^{1/\lambda}$$

$$m_{\text{b}}(x_t^{\text{pa}(i)})^{1/\lambda} = \bar{m}_{\text{b}}(x_t^{\text{pa}(i)}) = \left( \sum_a (\bar{Q}(x_t^{\text{pa}(i)}, a_t) \bar{n}^{(i)}(a_t))^{\frac{1}{\alpha}} \right)^\alpha$$

$$m^{(i)}(a_t)^{1/\lambda} = \bar{m}^{(i)}(a_t) = \left( \sum_{x_t^{\text{pa}(i)}} \left( \frac{\bar{Q}(x_t^{\text{pa}(i)}, a_t)}{\bar{m}_{\text{b}}(x_t^{\text{pa}(i)})} \right)^{\frac{1}{\alpha}} m_{\text{f}}(x_t^{\text{pa}(i)}) \bar{m}_{\text{b}}(x_t^{\text{pa}(i)})^\lambda \right)^\alpha$$

$$n^{(i)}(a_t)^{1/\lambda} = \bar{n}^{(i)}(a_t) = \prod_{k \neq i} \bar{m}^{(k)}(a_t)$$

$$m_{\text{f}}(x_{t+1}^{(i)}) = \sum_{x_t^{\text{pa}(i)}, a} \left( \frac{\bar{Q}(x_t^{\text{pa}(i)}, a_t) \bar{n}^{(i)}(a_t)}{\bar{m}_{\text{b}}(x_t^{\text{pa}(i)})} \right)^{\frac{1}{\alpha}} m_{\text{f}}(x_t^{\text{pa}(i)}) \bar{m}_{\text{b}}(x_t^{\text{pa}(i)})^\lambda \frac{p(x_{t+1}^{(i)}|x_t^{\text{pa}(i)}, a_t)}{\bar{Q}(x_t^{\text{pa}(i)}, a_t)^\lambda}$$

$$m_{\text{f}}(x_t^{\text{pa}(i)}) = \prod_{k \in \text{pa}(i)} n_{\text{f}}^{(i)}(x_t^{(k)})$$

$$n_{\text{f}}^{(i)}(x_t^{(j)}) = m_{\text{f}}(x_t^{(j)}) \prod_{k|j \in \text{pa}(k), k \neq i} \bar{n}_{\text{b}}^{(k)}(x_t^{(j)})^\lambda \quad \text{towards parents of entity } i$$

$$n_{\text{b}}^{(i)}(x_t^{(j)})^{1/\lambda} = \bar{n}_{\text{b}}^{(i)}(x_t^{(j)}) = \left( \sum_{\{x_t^{(k)}\}_{k \neq j}} \bar{m}_{\text{b}}(x_t^{\text{pa}(i)})^\lambda \prod_{k \neq i} n_{\text{f}}^{(i)}(x_t^{(k)}) \right)^{1/\lambda} \quad \text{from parents of entity } i$$

$$m_{\text{b}}(x_t^{(j)})^{1/\lambda} = \bar{m}_{\text{b}}(x_t^{(j)}) = \prod_{k|j \in \text{pa}(k)} \bar{n}_{\text{b}}^{(k)}(x_t^{(j)})$$

The following messages should be held constant

$$\bar{m}_{\text{b}}(x_t^{\text{pa}(i)}) = \exp(R(x_t^{\text{pa}(i)})) \;\; \forall i > N_e$$
$$m_{\text{f}}(x_1^{(i)}) = P(x_1^{(i)})$$

It is trivial to obtain these updates from Appendix D and setting the smoothing $\epsilon = \alpha\lambda$. This message updates correspond to maximum-entropy VBP, where $\lambda$ controls the exponential utility (from additive to multiplicative and beyond) and $\alpha$ the degree of policy entropy regularization.

### E.3 MaxEnt VBP with additive rewards (the $\lambda \to 0^+$ case)

If we take the limit $\lambda \to 0^+$ in the previous updates, we are specializing them to the additive rewards case, which is the standard setting in planning and reinforcement learning, while still keeping a parameter $\alpha$ that controls the degree of policy entropy regularization. When the graphical model is a non-factored MDP, MaxEnt VBP with $\lambda \to 0^+$ coincides with the dynamical programming updates proposed in (Levine, 2018, Section 3).

Thus, the updates below can be seen as an extension of the dynamic programming approach from (Levine, 2018) to factored MDPs:

$$\bar{Q}(x_t^{\mathrm{pa}(i)}, a_t) = \exp\Big( \sum_{x_{t+1}^{(i)}} p(x_{t+1}^{(i)}|x_t^{\mathrm{pa}(i)}, a_t) \log \bar{m}_{\mathrm{b}}(x_{t+1}^{(i)}) \Big)$$

$$\bar{m}_{\mathrm{b}}(x_t^{\mathrm{pa}(i)}) = \Big( \sum_a (\bar{Q}(x_t^{\mathrm{pa}(i)}, a_t)\bar{n}^{(i)}(a_t))^{\frac{1}{\alpha}} \Big)^{\alpha}$$

$$\bar{m}^{(i)}(a_t) = \Big( \sum_{x_t^{\mathrm{pa}(i)}} \Big( \frac{\bar{Q}(x_t^{\mathrm{pa}(i)}, a_t)}{\bar{m}_{\mathrm{b}}(x_t^{\mathrm{pa}(i)})} \Big)^{\frac{1}{\alpha}} m_{\mathrm{f}}(x_t^{\mathrm{pa}(i)}) \Big)^{\alpha}$$

$$\bar{n}^{(i)}(a_t) = \prod_{k \neq i} \bar{m}^{(k)}(a_t)$$

$$m_{\mathrm{f}}(x_{t+1}^{(i)}) = \sum_{x_t^{\mathrm{pa}(i)}, a} \Big( \frac{\bar{Q}(x_t^{\mathrm{pa}(i)}, a_t)\bar{n}^{(i)}(a_t)}{\bar{m}_{\mathrm{b}}(x_t^{\mathrm{pa}(i)})} \Big)^{\frac{1}{\alpha}} m_{\mathrm{f}}(x_t^{\mathrm{pa}(i)})p(x_{t+1}^{(i)}|x_t^{\mathrm{pa}(i)}, a_t)$$

$$m_{\mathrm{f}}(x_t^{\mathrm{pa}(i)}) = \prod_{k \in \mathrm{pa}(i)} n_{\mathrm{f}}^{(i)}(x_t^{(k)})$$

$$\bar{n}_{\mathrm{b}}^{(i)}(x_t^{(j)}) = \exp\Big( \sum_{\{x_t^{(k)}\}_{k \neq j}} (\log \bar{m}_{\mathrm{b}}(x_t^{\mathrm{pa}(i)})) \prod_{k \neq i} m_{\mathrm{f}}(x_t^{(k)}) \Big) \quad \text{from parents of entity } i$$

$$\bar{m}_{\mathrm{b}}(x_t^{(j)}) = \prod_{k | j \in \mathrm{pa}(k)} \bar{n}_{\mathrm{b}}^{(k)}(x_t^{(j)})$$

The following messages should be held constant

$$\bar{m}_{\mathrm{b}}(x_t^{\mathrm{pa}(i)}) = \exp(R(x_t^{\mathrm{pa}(i)})) \ \forall i > N_e$$
$$m_{\mathrm{f}}(x_1^{(i)}) = P(x_1^{(i)})$$

These are obtained directly from the ones in the previous subsection, simply by taking the limit $\lambda \to 0^+$, which remains well-defined.

### E.4 Empirical results using MaxEnt VBP with additive rewards

We provide here a repetition of the IPPC experiments from Section 6, but this time using MaxEnt VBP with additive rewards ($\lambda \to 0^+$). Results are similar, but the logic of the message passing is greatly simplified due to $\lambda$ having vanished analytically rather than numerically. The corresponding code is also included at `https://github.com/google-deepmind/what_type_of_inference_is_planning`.

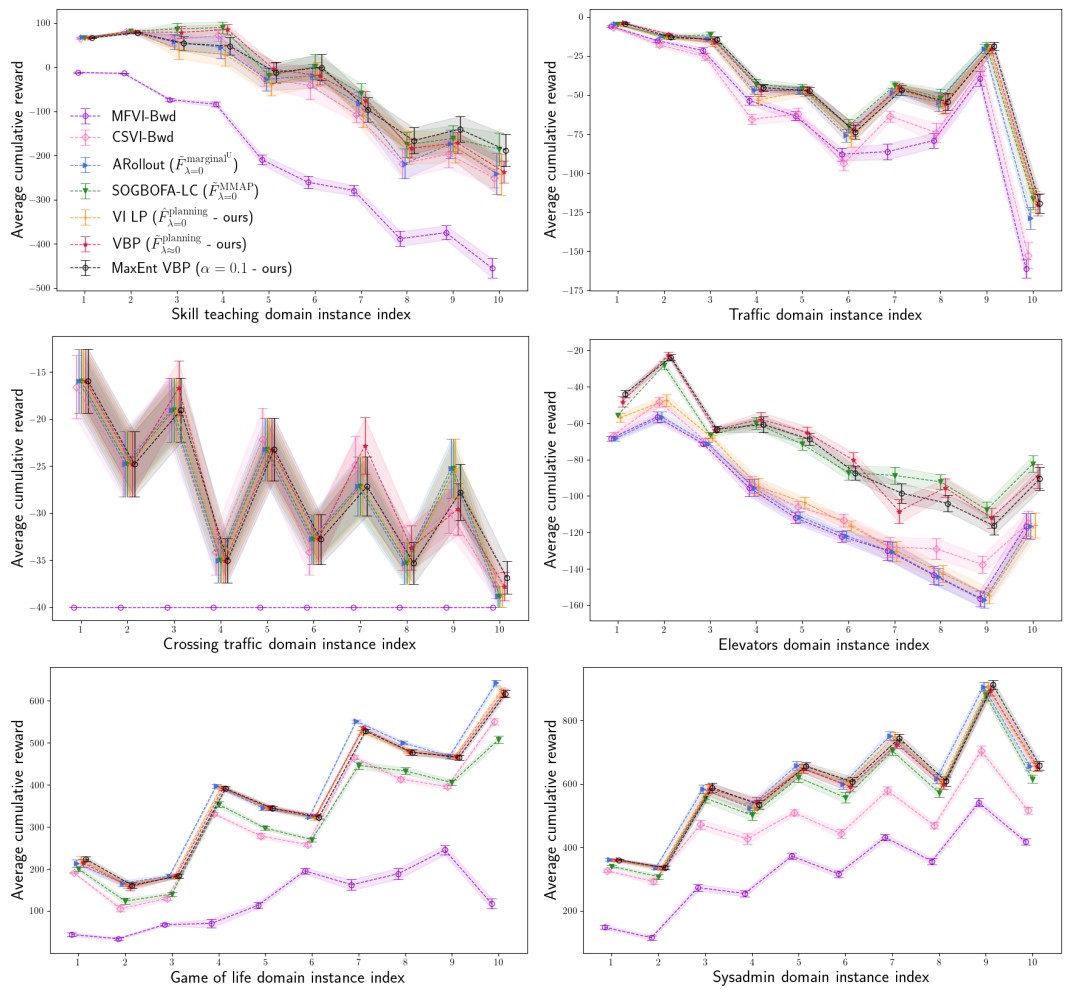

Figure 5: Cumulative rewards on 6 problem domains from the ICAPS 2011 IPPC. A small horizontal jitter was introduced in all data points for visual clarity. Each cumulative reward is averaged over 30 simulations per instance. Datasets are ordered from left to right and top to bottom by increasing normalized entropy levels. Only the last two have a significant stochasticity level >5%.

## F  Entropy terms for other inference types

The entropy terms for inference types other than the "planning inference" were introduced in prior work and are compiled here for convenience.

First we will derive a general variational expression that depends on an inverse temperature $\beta$. The function $f(\boldsymbol{x}, \boldsymbol{a})$ will be an (unnormalized) factor graph, with the structure of Fig. 1[Left].

$$\frac{1}{\beta} \log \sum_{\boldsymbol{x}, \boldsymbol{a}} f(\boldsymbol{x}, \boldsymbol{a})^\beta = \frac{1}{\beta} \log \sum_{\boldsymbol{x}, \boldsymbol{a}} q(\boldsymbol{x}, \boldsymbol{a}) \frac{f(\boldsymbol{x}, \boldsymbol{a})^\beta}{q(\boldsymbol{x}, \boldsymbol{a})} \stackrel{\text{tight Jensen}}{=} \max_{q(\boldsymbol{x}, \boldsymbol{a})} \frac{1}{\beta} \sum_{\boldsymbol{x}, \boldsymbol{a}} q(\boldsymbol{x}, \boldsymbol{a}) \log \frac{f(\boldsymbol{x}, \boldsymbol{a})^\beta}{q(\boldsymbol{x}, \boldsymbol{a})}$$

$$= \max_{q(\boldsymbol{x}, \boldsymbol{a})} \langle \log f(\boldsymbol{x}, \boldsymbol{a}) \rangle_{q(\boldsymbol{x}, \boldsymbol{a})} + \frac{1}{\beta} H_q(\boldsymbol{x}, \boldsymbol{a}) \tag{12}$$

**Marginal inference**  This is the standard VI problem, see e.g., (Jordan et al., 1999), and can be recovered from Eq. (12) by setting $\beta = 1$. Then we get

$$\log \sum_{\boldsymbol{x}, \boldsymbol{a}} f(\boldsymbol{x}, \boldsymbol{a}) = \max_{q(\boldsymbol{x}, \boldsymbol{a})} \langle \log f(\boldsymbol{x}, \boldsymbol{a}) \rangle_{q(\boldsymbol{x}, \boldsymbol{a})} + H_q(\boldsymbol{x}, \boldsymbol{a}). \tag{13}$$

Since $f(\boldsymbol{x}, \boldsymbol{a})$ forms a chain in our application of interest (see Fig. 1[Left]), we know that the optimal posterior will as well. We can thus decompose $H_q(\boldsymbol{x}, \boldsymbol{a})$ using the chain rule to obtain the expression from Table 1, $H^{\text{marginal}}(\boldsymbol{q}) = H_q(x_1) + \sum_{t=1}^{T-1} H_q(x_{t+1}, a_t | x_t)$.

**Maximum-a-posterior (MAP) inference** MAP inference can be recovered from Eq. (12) by setting $\beta \to \infty$

$$\max_{\boldsymbol{x}, \boldsymbol{a}} \log f(\boldsymbol{x}, \boldsymbol{a}) = \lim_{\beta \to \infty} \frac{1}{\beta} \log \sum_{\boldsymbol{x}, \boldsymbol{a}} f(\boldsymbol{x}, \boldsymbol{a})^\beta = \max_{q(\boldsymbol{x}, \boldsymbol{a})} \langle \log f(\boldsymbol{x}, \boldsymbol{a}) \rangle_{q(\boldsymbol{x}, \boldsymbol{a})} + \lim_{\beta \to \infty} \frac{1}{\beta} H_q(\boldsymbol{x}, \boldsymbol{a}), \quad (14)$$

where $\lim_{\beta \to \infty} \frac{1}{\beta} H_q(\boldsymbol{x}, \boldsymbol{a})$ produces the term $H^{\text{MAP}}(\boldsymbol{q}) = 0$ from Table 1. The maximization is an LP problem, see (Weiss et al., 2012). This problem can be relaxed into a local polytope, giving rise to most well-known methods for approximate MAP inference, such as dual decomposition. See e.g., (Sontag et al., 2011). An optimal distribution solving this problem is a Dirac delta centered at one of the MAP solutions of the problem.

**Marginal maximum-a-posterior (MMAP) inference** MMAP combines the previous two types of inference, see (Liu and Ihler, 2013) for detailed (approximate) solution methods. We can reuse the previous results to obtain the variational expression

$$\max_{\boldsymbol{a}} \log \sum_{\boldsymbol{x}} f(\boldsymbol{x}, \boldsymbol{a}) \overset{\text{Eq. (13)}}{=} \max_{\boldsymbol{a}} \max_{q(\boldsymbol{x}|\boldsymbol{a})} \langle \log f(\boldsymbol{x}, \boldsymbol{a}) \rangle_{q(\boldsymbol{x}|\boldsymbol{a})} + H(q(\boldsymbol{x}|\boldsymbol{a}))$$

$$\overset{\text{Eq. (14)}}{=} \max_{q(\boldsymbol{x}, \boldsymbol{a})} \langle \log f(\boldsymbol{x}, \boldsymbol{a}) \rangle_{q(\boldsymbol{x}, \boldsymbol{a})} + H_q(\boldsymbol{x}, \boldsymbol{a}) - H_q(\boldsymbol{a})$$

$$= \max_{q(\boldsymbol{x}, \boldsymbol{a})} \langle \log f(\boldsymbol{x}, \boldsymbol{a}) \rangle_{q(\boldsymbol{x}, \boldsymbol{a})} + H_q(\boldsymbol{x}, \boldsymbol{a}) - \sum_{t=1}^{T-1} H_q(a_t)$$

To understand the last equality, first note that $\sum_{t=1}^{T-1} H_q(a_t) \geq H_q(\boldsymbol{a})$, since the latter is the joint entropy. This means that the last expression is necessarily lower than or equal to the previous one. It is not yet clear why equality is always achievable. To see this note that, analogously to the previous section, the optimal $q(\boldsymbol{a})$ will be a Dirac delta centered at an optimal sequence $\boldsymbol{a}$. That means that the optimal variational distribution of MMAP factorizes as $q(\boldsymbol{x}, \boldsymbol{a}) = q(\boldsymbol{x}) \prod_{t=1}^{T-1} q(a_t)$. This in turn means that, for the optimal distribution $\sum_{t=1}^{T-1} H_q(a_t) = H_q(\boldsymbol{a})$. So the last expression can always match the previous one, and given that is also a lower bound, it must have the same optimal distribution and value.

Just like we did for marginal inference, we can now decompose $H_q(\boldsymbol{x}, \boldsymbol{a})$ using the chain rule to obtain the expression from Table 1, $H^{\text{MMAP}}(\boldsymbol{q}) = H_q(x_1) + \sum_{t=1}^{T-1} H_q(x_{t+1}, a_t | x_t) - H_q(a_t)$.

## G  Proof of the bounding relationships among different inference types

We want to prove that for an arbitrary variational distribution $\boldsymbol{q}$

$$\left. \begin{array}{c} F_\lambda^{\text{MAP}}(\boldsymbol{q}) \\ F_\lambda^{\text{marginal}^{\text{U}}}(\boldsymbol{q}) \end{array} \right\} \leq F_\lambda^{\text{MMAP}}(\boldsymbol{q}) \leq F_\lambda^{\text{planning}}(\boldsymbol{q}) \leq F_\lambda^{\text{marginal}}(\boldsymbol{q})$$

and that, when we assume that dynamics are deterministic (which we also take to imply that the first state follows a deterministic distribution, i.e., it is known), the optimal values of the above bounds wrt $\boldsymbol{q}$ satisfy the following relationships

$$F_\lambda^{\text{marginal}^{\text{U}}} \leq F_\lambda^{\text{MAP}} = F_\lambda^{\text{MMAP}} = F_\lambda^{\text{planning}} \leq F_\lambda^{\text{marginal}}.$$

Since the energy terms are identical for all these bounds, we will simply have to prove the relationship between the corresponding entropies, as given in Table 1.

**Proof that $F_\lambda^{\text{planning}}(\boldsymbol{q}) \leq F_\lambda^{\text{marginal}}(\boldsymbol{q})$**

The marginal entropy term

$$H^{\text{marginal}}(\boldsymbol{q}) = H_q(x_1) + \sum_{t=1}^{T-1} H_q(x_{t+1}, a_t | x_t)$$

clearly upper bounds the "planning inference" entropy term

$$H^{\text{planning}}(\boldsymbol{q}) = H_q(x_1) + \sum_{t=1}^{T-1} H_q(x_{t+1} | a_t, x_t)$$

since

$$H_q(x_{t+1}, a_t | x_t) = H_q(x_{t+1} | a_t, x_t) + H_q(a_t | x_t),$$

and entropies are non-negative.

**Proof that $F_\lambda^{\text{MMAP}}(\boldsymbol{q}) \leq F_\lambda^{\text{planning}}(\boldsymbol{q})$ (and $F_\lambda^{\text{MMAP}} = F_\lambda^{\text{planning}}$ for deterministic dynamics)**

The "planning inference" entropy term

$$H^{\text{planning}}(\boldsymbol{q}) = H_q(x_1) + \sum_{t=1}^{T-1} H_q(x_{t+1} | a_t, x_t)$$

clearly upper bounds the MMAP entropy term

$$\begin{aligned}
H^{\text{MMAP}}(\boldsymbol{q}) &= H_q(x_1) + \sum_{t=1}^{T-1} H_q(x_{t+1}, a_t | x_t) - H_q(a_t) \\
&= H_q(x_1) + \sum_{t=1}^{T-1} H_q(x_{t+1} | x_t, a_t) + (H_q(a_t | x_t) - H_q(a_t)) \\
&= H_q(x_1) + \sum_{t=1}^{T-1} H_q(x_{t+1} | x_t, a_t) - I_q(x_t; a_t)
\end{aligned}$$

since the mutual information $I_q(x_t; a_t)$ is non-negative.

In exact MMAP variational inference (Liu and Ihler, 2013), at the optimal solution, the variational distribution factorizes as $q(\boldsymbol{x}, \boldsymbol{a}) = q(\boldsymbol{x}) \prod_{t=1}^{T-1} q(a_t)$, as we mentioned in Appendix F. This is because MMAP finds a single, deterministic, optimal sequence of actions. Therefore $I_q(x_t; a_t) = 0$ when evaluated at the $\boldsymbol{q}$ that maximizes $F_\lambda^{\text{MMAP}}(\boldsymbol{q})$. Setting that term to zero in the MMAP entropy results in the same expression as the "planning inference" entropy. However, in "planning inference", the optimal variational distribution does not need to factorize in the way it does for MMAP, so a richer variational distribution is possible, and $F_\lambda^{\text{planning}}$ can be strictly larger than $F_\lambda^{\text{MMAP}}$ for some problems.

However, when dynamics are deterministic and the first state $x_1$ is known, the optimal plan is a deterministic sequence of states and actions, and the optimal "planning" variational distribution is a Dirac delta at those states and actions. Therefore, the optimal distribution for planning also factorizes as $q(\boldsymbol{x}, \boldsymbol{a}) = q(\boldsymbol{x}) \prod_{t=1}^{T-1} q(a_t)$ when the dynamics are deterministic. Therefore, in that case the optimal values coincide $F_\lambda^{\text{planning}} = F_\lambda^{\text{MMAP}}$ (and so do the optimal variational distributions).

We can also show that at the optimal $\boldsymbol{q}$ value the entropies of both bounds are zero. $P(x_{t+1} | a_t, x_t)$ (and $P(x_1)$) are deterministic, $q(x_{t+1} | a_t, x_t) = P(x_{t+1} | a_t, x_t)$ (and $q(x_1) = P(x_1)$) for both of these bounds to be larger than $-\infty$ (see Section 4.2). Since these are deterministic distributions, $H_q(x_{t+1} | a_t, x_t) = 0$ and $H_q(x_1) = 0$, which in turn makes $H^{\text{planning}}(\boldsymbol{q}) = 0$ at the optimal $\boldsymbol{q}$. I.e., $H^{\text{planning}} = 0$. Since $I_q(x_t; a_t) = 0$, also $H^{\text{MMAP}} = 0$.

**Proof that $F_\lambda^{\text{MAP}}(\boldsymbol{q}) \leq F_\lambda^{\text{MMAP}}$ (and $F_\lambda^{\text{MAP}} = F_\lambda^{\text{MMAP}}(\boldsymbol{q})$ for deterministic dynamics)**

The MMAP entropy can be rearranged in the following form

$$H^{\text{MMAP}}(\boldsymbol{q}) = H_q(x_1) + \sum_{t=1}^{T-1} H_q(x_{t+1}, a_t | x_t) - H_q(a_t)$$

$$= H_q(x_1, a_1, \ldots, x_{T-1}, a_{T-1}, x_T) - \sum_{t=1}^{T-1} H_q(a_t)$$

$$= I_q(x_1; a_1; \ldots; x_{T-1}; a_{T-1}; x_T) + \sum_{t=1}^{T} H_q(x_t) \geq 0,$$

where the last equality follows because the mutual information and entropy are non-negative. This clearly upper bounds the MAP entropy term $H^{\text{MAP}}(\boldsymbol{q}) = 0$.

When dynamics are deterministic and the first state $x_1$ is known, we can reuse the results of the previous proof. We know that in that case, the optimal variational distribution for MMAP and planning is a Dirac delta at the optimal sequence of states and actions, and that the bound only contains the energy terms (since the entropy terms for any Dirac delta distribution will be zero). The problem of finding such distribution is exactly the MAP energy minimization problem. Thus, if dynamics are deterministic the optimal values coincide $F_\lambda^{\text{MMAP}} = F_\lambda^{\text{MAP}}$ (and so do the optimal variational distributions).

**Proof that $F_\lambda^{\text{marginal}^{\text{U}}}(\boldsymbol{q}) \leq F_\lambda^{\text{MMAP}}(\boldsymbol{q})$**

The MMAP entropy term

$$H^{\text{MMAP}}(\boldsymbol{q}) = H_q(x_1) + \sum_{t=1}^{T-1} (H_q(x_{t+1}, a_t | x_t) - H_q(a_t))$$

clearly upper bounds $H^{\text{Marginal}^{\text{U}}}(\boldsymbol{q})$, i.e.,

$$H^{\text{Marginal}^{\text{U}}}(\boldsymbol{q}) = H_q(x_1) + \sum_{t=1}^{T-1} (H_q(x_{t+1}, a_t | x_t) - \log N_a)$$

since $H_q(a_t) \leq \log N_a$, with equality being attained when $q(a_t)$ is uniform.

**Proof that $F_\lambda^{\text{marginal}^{\text{U}}} \leq F_\lambda^{\text{MAP}}$ for deterministic dynamics**

We can rewrite the entropy corresponding to the marginal bound with uniform prior as

$$H^{\text{Marginal}^{\text{U}}}(\boldsymbol{q}) = H_q(x_1) + \sum_{t=1}^{T-1} (H_q(x_{t+1}, a_t | x_t) - \log N_a)$$

$$= H_q(x_1) + \sum_{t=1}^{T-1} (H_q(x_{t+1} | a_t, x_t) + H_q(a_t | x_t) - \log N_a).$$

When dynamics are deterministic and the first state $x_1$ is known, we know from the previous proofs that at the optimal value $\boldsymbol{q}$ of $F_\lambda^{\text{marginal}^{\text{U}}}(\boldsymbol{q})$ we have $H_q(x_1) = 0$ and $H_q(x_{t+1} | a_t, x_t) = 0$. The remaining terms, of the form $H_q(a_t | x_t) - \log N_a$ are trivially non-positive (the conditional entropy over the actions cannot be larger than the log of the cardinality of the action space). This means that at the maximum value of $F_\lambda^{\text{marginal}^{\text{U}}}(\boldsymbol{q})$ wrt $\boldsymbol{q}$ we have $H^{\text{Marginal}^{\text{U}}}(\boldsymbol{q}) \leq 0$. Since $H^{\text{MAP}}(\boldsymbol{q}) = 0$, we know that with deterministic dynamics $F_\lambda^{\text{marginal}^{\text{U}}} \leq F_\lambda^{\text{MAP}}$.

# H  An application of VI for planning: bounding determinization in hindsight

In the planning literature, many algorithms make the assumption of a deterministic environment (Geffner and Bonet, 2022; Hoffmann and Nebel, 2001; Helmert, 2006; etc). In order to extend these algorithms to the case of stochastic dynamics, the idea of *determinization in hindsight* has

been developed (Yoon et al., 2008). It starts by factoring out all the stochasticity from the transition probability into the random variables $\boldsymbol{\gamma} = \{\gamma_t\}_{t=1}^{t=T-1}$

$$P(x_{t+1}|x_t, a_t) = \sum_{\gamma_t} P_{\text{det}}(x_{t+1}|x_t, a_t, \gamma_t)P(\gamma_t) = \langle P_{\text{det}}(x_{t+1}|x_t, a_t, \gamma_t)\rangle_{P(\gamma_t)}.$$

This factorization is always possible, and $\boldsymbol{\gamma}$ is known as the collection of exogenous variables in structural equation modeling (Pearl, 2012; Bareinboim and Pearl, 2016; Rubenstein et al., 2017). We will define $P_{\text{det}}(\boldsymbol{x}|\boldsymbol{a}, \boldsymbol{\gamma}) \equiv P_{\text{det}}(x_1|\gamma_0)\prod_{t=1}^{T-1} P_{\text{det}}(x_{t+1}|x_t, a_t, \gamma_t)$. Then we can use it to obtain an upper bound on the exact utility

$$F_{\lambda=1}^{\text{planning}} = \log \max_{\boldsymbol{\pi}}\langle\langle\exp(R(\boldsymbol{x}, \boldsymbol{a}))\rangle_{P_{\text{det}}(\boldsymbol{x}|\boldsymbol{a}, \boldsymbol{\gamma})\pi(\boldsymbol{a}|\boldsymbol{x})}\rangle_{P(\boldsymbol{\gamma})}$$

$$\leq \log\langle\max_{\boldsymbol{\pi}}\langle\exp(R(\boldsymbol{x}, \boldsymbol{a}))\rangle_{P_{\text{det}}(\boldsymbol{x}|\boldsymbol{a}, \boldsymbol{\gamma})\pi(\boldsymbol{a}|\boldsymbol{x})}\rangle_{P(\boldsymbol{\gamma})} = F_{\lambda=1}^{\text{det. planning}}$$

that has a natural interpretation: the dynamics of a stochastic environment depends on random variables $\boldsymbol{\gamma}$ that get drawn at each time step, if we knew their value ahead of time, we could treat the environment as deterministic, find an optimal plan and estimate its utility. Because we have *hindsight*, i.e., the deterministic planner can see the value of future $\gamma_t$ ahead of time, it can make better choices than if these were revealed online, so the utility estimation, if exact, upper bounds the original quantity. See (Yoon et al., 2008) for more details.

### H.1 Upper bounding determinization for standard MDPs

The above process uses a sample average to compute the determinization objective: it alternates between sampling $\boldsymbol{\gamma} \sim P(\boldsymbol{\gamma})$ and running a deterministic planner, with the exact value being attainable only in the infinite limit. Using the variational framework for planning it is possible to find and upper bound for $F_{\lambda=1}^{\text{det. planning}}$ as the maximum of a concave function that is computable in polynomial time.

First, let us expand the determinization bound

$$F_{\lambda=1}^{\text{det. planning}} \overset{\text{Eq. (9)}}{=} \max_{q(\boldsymbol{\gamma})}\langle\log\max_{\boldsymbol{\pi}}\langle\exp(R(\boldsymbol{x}, \boldsymbol{a}))\rangle_{P_{\text{det}}(\boldsymbol{x}|\boldsymbol{a}, \boldsymbol{\gamma})\pi(\boldsymbol{a}|\boldsymbol{x})} + \log P(\boldsymbol{\gamma})\rangle_{q(\boldsymbol{\gamma})} + H_q(\boldsymbol{\gamma})$$

$$= \max_{q(\boldsymbol{\gamma})}\langle F_{\lambda=1, \boldsymbol{\gamma}}^{\text{planning}}\rangle_{q(\boldsymbol{\gamma})} + \langle\log P(\boldsymbol{\gamma})\rangle_{q(\boldsymbol{\gamma})} + H_q(\boldsymbol{\gamma})$$

in terms of our exact planning utility for a given future $\boldsymbol{\gamma}$, $F_{\lambda=1, \boldsymbol{\gamma}}^{\text{planning}}$. This quantity and the corresponding bound are defined in the usual way, but conditioned on $\boldsymbol{\gamma}$:

$$F_{\lambda=1, \boldsymbol{\gamma}}^{\text{planning}} = \max_{\boldsymbol{q}} F_{\lambda=1}^{\text{planning}}(\boldsymbol{q}|\boldsymbol{\gamma})$$

with

$$F_{\lambda=1}^{\text{planning}}(\boldsymbol{q}|\boldsymbol{\gamma}) = \langle\log P_{\text{det}}(x_1|\gamma_0)\rangle_{q(x_1|\boldsymbol{\gamma})}$$

$$+ \sum_{t=1}^{T-1}\langle R_t(x_t, a_t, x_{t+1}) + \log P_{\text{det}}(x_{t+1}|x_t, a_t, \gamma_t)\rangle_{q(x_{t+1}, x_t, a_t|\boldsymbol{\gamma})}. \quad (15)$$

Since the conditional dynamics are now deterministic, $\log P_{\text{det}}(x_{t+1}|x_t, a_t, \gamma_t)$ can only take the values 0 and $-\infty$. When maximizing over $\boldsymbol{q}$, it is obvious we must choose $q(x_{t+1}|x_t, a_t, \boldsymbol{\gamma}) = P_{\text{det}}(x_{t+1}|x_t, a_t, \gamma_t) \,\forall t$ and $q(x_1|\boldsymbol{\gamma}) = P_{\text{det}}(x_1|\gamma_0)$, to avoid putting any mass on the $-\infty$ value of the previous term, which would result in the whole score function becoming $-\infty$. For such choice the planning entropy vanishes, since dynamics are deterministic.

The expectation $\langle F_{\lambda=1}^{\text{planning}}(\boldsymbol{q}|\boldsymbol{\gamma})\rangle_{q(\boldsymbol{\gamma})}$ initially seems to depend on the entire distribution $q(\boldsymbol{\gamma})$. However, only the marginals $q(\gamma_t)$ are actually relevant for its computation:

$$
\begin{aligned}
\langle F_{\lambda=1}^{\text{planning}}(\boldsymbol{q}|\boldsymbol{\gamma})\rangle_{q(\boldsymbol{\gamma})} =& \langle \log P_{\text{det}}(x_1|\gamma_0)\rangle_{q(x_1|\boldsymbol{\gamma})q(\boldsymbol{\gamma})} \\
& + \sum_{t=1}^{T-1} \langle R_t(x_t, a_t, x_{t+1}) + \log P_{\text{det}}(x_{t+1}|x_t, a_t, \gamma_t)\rangle_{q(x_{t+1},x_t,a_t|\boldsymbol{\gamma})q(\boldsymbol{\gamma})} \\
=& \langle \log P_{\text{det}}(x_1|\gamma_0)\rangle_{q(x_1|\gamma_0)q(\gamma_0)} \\
& + \sum_{t=1}^{T-1} \langle R_t(x_t, a_t, x_{t+1}) + \log P_{\text{det}}(x_{t+1}|x_t, a_t, \gamma_t)\rangle_{q(x_{t+1},x_t,a_t|\gamma_t)q(\gamma_t)}
\end{aligned}
$$

So far all computations are exact. If we now upper bound the entropy of the exogenous variables, $H_q(\boldsymbol{\gamma}) \leq \sum_{t=0}^{T-1} H_q(\gamma_t)$, we can upper bound the determinization objective. Putting it all together, we have that *determinization for multiplicative exponentiated rewards can be upper bounded by a concave variational bound of the form*

$$
F_{\lambda=1}^{\text{det. planning}} \leq F_{\lambda=1}^{\text{det. planning UB}} = \max_{\boldsymbol{q}_\gamma} F_{\lambda=1}^{\text{planning}}(\boldsymbol{q}_\gamma) + \sum_{t=0}^{T-1} \langle \log P(\gamma_t)\rangle_{q(\gamma_t)} + H_q(\gamma_t) \tag{16}
$$

$$
\begin{aligned}
\text{s.t.} \quad & \sum_{a_1,\gamma_1} q(x_1, a_1, \gamma_1) = \sum_{\gamma_0} P_{\text{det}}(x_1|\gamma_0)q(\gamma_0) \\
& \sum_{a_{t+1},\gamma_{t+1}} q(x_{t+1}, a_{t+1}, \gamma_{t+1}) = \sum_{x_t,a_t,\gamma_t} P_{\text{det}}(x_{t+1}|x_t, a_t, \gamma_t)q(x_t, a_t, \gamma_t) \;\; \forall t; \\
& q(x_t, \gamma_t) = \sum_{a_t} q(x_t, a_t, \gamma_t) \;\; \forall t; \quad q(x_t, a_t, \gamma_t) \geq 0 \;\forall t,
\end{aligned}
$$

where we have defined

$$
F_{\lambda=1}^{\text{planning}}(\boldsymbol{q}_\gamma) = \sum_{t=1}^{T-1} \langle R_t(x_t, a_t, x_{t+1})\rangle_{P_{\text{det}}(x_{t+1}|x_t,a_t,\gamma_t)q(x_t,a_t,\gamma_t)}
$$

as a simplification of $\langle F_{\lambda=1}^{\text{planning}}(\boldsymbol{q}|\boldsymbol{\gamma})\rangle_{q(\boldsymbol{\gamma})}$ when the above constraints are met. We have additionally defined $\boldsymbol{q}_\gamma \equiv \{q(x_t, a_t, \gamma_t)\}_{t=1}^{T-1} \cup q(\gamma_0)$.

## H.2 The factored MDP case

In the case of a standard MDP, it is trivial to see that determinization, which provides an upper bound on the exact utility, cannot improve on our VI bound $F_{\lambda=1}^{\text{planning}}$, since the latter is exact. In this section we will show that this is also the case for factored MDPs in which the deterministic MAP problem is solved using an LP MAP relaxation.

First, we will show that $F_{\lambda=1}^{\text{planning}}$ can be rewritten in a more convenient way to compare with determinization, in the standard, non-factored case. We expand

$$
F_{\lambda=1}^{\text{planning}} = \max_{\boldsymbol{q}} F_{\lambda=1}^{\text{planning}}(\boldsymbol{q}) = \max_{\boldsymbol{q}} -E_{\lambda=1}(\boldsymbol{q}) + H^{\text{planning}}(\boldsymbol{q})
$$

and substitute

$$
\begin{aligned}
\log P(x_{t+1}|x_t, a_t) = \max_{q(\gamma_t|x_{t+1},x_t,a_t)} \Big( & \langle \log P_{\text{det}}(x_{t+1}|x_t, a_t, \gamma_t)\rangle_{q(\gamma_t|x_{t+1},x_t,a_t)} \\
& - \sum_{t=1}^{T} \text{KL}(q(\gamma_t|x_{t+1}, x_t, a_t)||P(\gamma_t)) \Big)
\end{aligned}
$$

inside $E_{\lambda=1}(\boldsymbol{q})$ to get

$$
F_{\lambda=1}^{\text{planning}} = \max_{\boldsymbol{q}} \langle F_{\lambda=1}^{\text{planning}}(\boldsymbol{q}|\boldsymbol{\gamma})\rangle_{q(\boldsymbol{\gamma})} + \sum_{t=0}^{T-1} \langle \log P(\gamma_t)\rangle_{q(\gamma_t)} + H_q(x_{t+1}, \gamma_t|a_t, x_t)
$$

where $q(x_{t+1}|x_t, a_t, \gamma_t) = P_{\text{det}}(x_{t+1}|x_t, a_t, \gamma_t) \ \forall t$ for the same reasons as above. To compact notation, we use the convention that $H_q(x_1, \gamma_0|x_0, a_0) = H_q(x_1, \gamma_0)$ (since $x_0, a_0$ are not defined), and similarly $H_q(\gamma_0|x_0, a_0) = H_q(\gamma_0)$. Noting that $H_q(x_{t+1}, \gamma_t|a_t, x_t) = H_q(x_{t+1}|x_t, a_t, \gamma_t) + H_q(\gamma_t|a_t, x_t)$ and that $H_q(x_{t+1}|x_t, a_t, \gamma_t) = 0$ because $q(x_{t+1}|x_t, a_t, \gamma_t)$ is deterministic, we get

$$F_{\lambda=1}^{\text{planning}} = \max_{\boldsymbol{q}_\gamma} F_{\lambda=1}^{\text{planning}}(\boldsymbol{q}_\gamma) + \sum_{t=0}^{T-1} \langle \log P(\gamma_t) \rangle_{q(\gamma_t)} + H_q(\gamma_t|a_t, x_t) \tag{17}$$

$$\text{s.t.} \sum_{a_1, \gamma_1} q(x_1, a_1, \gamma_1) = \sum_{\gamma_0} P_{\text{det}}(x_1|\gamma_0) q(\gamma_0)$$

$$\sum_{a_{t+1}, \gamma_{t+1}} q(x_{t+1}, a_{t+1}, \gamma_{t+1}) = \sum_{x_t, a_t, \gamma_t} P_{\text{det}}(x_{t+1}|x_t, a_t, \gamma_t) q(x_t, a_t, \gamma_t) \ \forall t;$$

$$q(x_t, \gamma_t) = \sum_{a_t} q(x_t, a_t, \gamma_t) \ \forall x_t, \gamma_t, t; \quad q(x_t, a_t, \gamma_t) \geq 0 \ \forall t.$$

Compare Eqs. (16) and (17). Although we already knew this, with this particular formulation it is very easy to see that $F_{\lambda=1}^{\text{planning}} \leq F_{\lambda=1}^{\text{det. planning UB}}$, since they are identical except for the terms $H_q(\gamma_t|a_t, x_t) \leq H_q(\gamma_t) \ \forall t$. This holds for the same $\boldsymbol{q}$, but also establishes a relationship between both upper bounds at their maximum wrt $\boldsymbol{q}$.

With this result, we can return to the factored MDP case. As in the main text, here we will consider $R$ to be only a function of $x_t$. We can take Eqs. (16) and (17) and relax $\boldsymbol{q}_\gamma$ from the marginal polytope $\mathcal{M}$ to the local polytope $\mathcal{L}$, using the pseudo-marginals $\tilde{\boldsymbol{q}}_\gamma \equiv \{q(x_t^{\text{pa}(i)}, a_t, \gamma_t^{(i)})\}_{t=1,i=1}^{t=T-1,i=N_e} \cup \{q(\gamma_0^{(i)})\}_{i=1}^{N_e} \cup \{q(\boldsymbol{x}_t^{\text{pa}(i)})\}_{t=1,i=N_e+1}^{t=T,i=N_e+N_r}$. For Eq. (16), and substituting the value from Eq. (15), this produces a factored MDP upper bound corresponding to determinization with $\lambda = 1$

$$\hat{F}_{\lambda=1}^{\text{det. planning UB}} = \max_{\tilde{\boldsymbol{q}}_\gamma} \sum_{t=1}^{T} \Big( \sum_{i=N_e+1}^{N_e+N_r} \langle R(\boldsymbol{x}_t^{\text{pa}(i)}) \rangle_{q(\boldsymbol{x}_t^{\text{pa}(i)})} + \sum_{i=1}^{N_e} \langle \log P(\gamma_{t-1}^{(i)}) \rangle_{q(\gamma_{t-1}^{(i)})} + H_q(\gamma_{t-1}^{(i)}) \Big)$$

s.t.

$$\sum_{a_1, \gamma_1^{(i)}} q(x_1^{(i)}, a_1, \gamma_1^{(i)}) = \sum_{\gamma_0^{(i)}} P_{\text{det}}(x_1^{(i)}|\gamma_0^{(i)}) q(\gamma_0^{(i)}) \ \forall i$$

$$\sum_{a_{t+1}, \gamma_{t+1}^{(i)}} q(x_{t+1}^{(i)}, a_{t+1}, \gamma_{t+1}^{(i)}) = \sum_{x_t^{\text{pa}(i)}, a_t, \gamma_t^{(i)}} P_{\text{det}}(x_{t+1}^{(i)}|x_t^{\text{pa}(i)}, a_t, \gamma_t^{(i)}) q(x_t^{\text{pa}(i)}, a_t, \gamma_t^{(i)}) \ \forall t, i \in 1, \ldots, N_e$$

and pseudo-marginal constraints,

which can be interpreted as an outer maximization over $q(\boldsymbol{\gamma})$ and an inner MAP optimization of the (conditional on $\boldsymbol{\gamma}$) deterministic dynamics problem, solved with an LP relaxation (Sontag et al., 2011). For Eq. (17), and also substituting the value from Eq. (15), this results in a different formulation of the problem Eq. (7)

$$\hat{F}_{\lambda=1}^{\text{planning}} = \max_{\tilde{\boldsymbol{q}}_\gamma} \sum_{t=1}^{T} \Big( \sum_{i=N_e+1}^{N_e+N_r} \langle R(\boldsymbol{x}_t^{\text{pa}(i)}) \rangle_{q(\boldsymbol{x}_t^{\text{pa}(i)})} + \sum_{i=1}^{N_e} \langle \log P(\gamma_{t-1}^{(i)}) \rangle_{q(\gamma_{t-1}^{(i)})} + H_q(\gamma_{t-1}^{(i)}|x_{t-1}, a_{t-1}) \Big)$$

s.t.

$$\sum_{a_1, \gamma_1^{(i)}} q(x_1^{(i)}, a_1, \gamma_1^{(i)}) = \sum_{\gamma_0^{(i)}} P_{\text{det}}(x_1^{(i)}|\gamma_0^{(i)}) q(\gamma_0^{(i)}) \ \forall i$$

$$\sum_{a_{t+1}, \gamma_{t+1}^{(i)}} q(x_{t+1}^{(i)}, a_{t+1}, \gamma_{t+1}^{(i)}) = \sum_{x_t^{\text{pa}(i)}, a_t, \gamma_t^{(i)}} P_{\text{det}}(x_{t+1}^{(i)}|x_t^{\text{pa}(i)}, a_t, \gamma_t^{(i)}) q(x_t^{\text{pa}(i)}, a_t, \gamma_t^{(i)}) \ \forall t, i \in 1, \ldots, N_e$$

and pseudo-marginal constraints.

In the same way as for the non-factorized MDP, it is easy to see that $\hat{F}_{\lambda=1}^{\text{planning}} \leq \hat{F}^{\text{det. planning UB}}$, while both upper bound the exact $F_{\lambda=1}^{\text{planning}}$. This means that *in the case of (exponentiated) multiplicative rewards ($\lambda = 1$) our proposed bound should be no worse than the provided upper bound on determinization.*

# I Empirical validation: details

## I.1 Synthetic MDPs

The synthetic MDPs use binary state and actions, $T = 4$ time steps and follow the connectivity scheme of Fig. 1[Right]. The stochasticity level of the MDPs is controlled by generating their dynamics according to $P(x_{t+1}|x_t, a_t) = \bar{P}^s_{x_{t+1},x_t,a_t}/Z_{x_t,a_t}$ where $\bar{P}_{x_{t+1},x_t,a_t} \sim U[0,1]$ and choosing the exponent $s$ and the divisor $Z_{x_t,a_t}$ to obtain normalized distributions of the desired total normalized entropy $H_{\text{MDP}}$. Each entity has two parent entities and we provide a single reward of 1 for reaching the state 0 of entity 1 at the last step $T$. Having a single reward allows comparing methods with arbitrary $\lambda$ and create a pure planning problem (see Section 4.2).

## I.2 Reactivity avoidance of MAP and MMAP inference

As discussed in Section 4.2, a common flaw among all inference types (other than planning inference) is that they cannot anticipate reacting to the environment. Even the tightest one, MMAP, makes plans assuming that it will only be able to take a predefined sequence of actions. This can severely underestimate the value of an action if, to extract future reward, reacting to the environment state is necessary. In other words, MMAP is optimal only when the best policy $\pi_t(a_t|x_t)$ ignores the state and can be represented as $\pi_t(a_t)$. MAP inference shares that limitation, and additionally lacks path integration. I.e., MAP inference makes decisions based on the score of a single action-state trajectory, rather than a combination of these.

In order to analyze the behavior of different inference types of inference with reactivity, we create an MDP with two entities, each with categorical states $0, \ldots, 5$. We use a horizon of $T = 7$ time steps, placing reward only on the last time step. There are a total of 8 actions.

The first entity describes the location of the agent, and has a special goal state "0" that needs to be reached at the last time step. The second entity describes the dynamics of the first entity and the reward achieved at the goal. Actions $0, \ldots, 5$ allow the agent to move to any location in a single step. Actions $6, 7$ allow the agent to modify the dynamics of the first entity, respectively decreasing or increasing the needed level of reactivity of a planner.

In more detail, the second entity has deterministic dynamics and acts as a knob that the agent can freely turn up or down at each time step:

$$x_{t+1}^{(2)} = \begin{cases} x_t^{(2)} - 1, & \text{if } a = 6 \text{ and } 0 < x_t^{(2)} \\ x_t^{(2)} + 1, & \text{if } a = 7 \text{ and } x_t^{(2)} < 5 \\ x_t^{(2)}, & \text{otherwise.} \end{cases}$$

The values of that knob $x_t^{(2)}$ alter the dynamics of the first entity, requiring more or less reactivity:

$$x_{t+1}^{(1)} = \begin{cases} \text{if } x_t^{(1)} = 0 \text{ or } 6 \leq a & \sim U[1,5] \\ \text{if } x_t^{(1)} \neq 0 \text{ and } 0 \leq a < 6 & \begin{cases} x_t^{(1)} + a \mod 6 & \text{with probability } x_t^{(2)}/5 \\ 0 & \text{with probability } 1 - x_t^{(2)}/5 \end{cases} \end{cases}$$

In words, if the agent is at the goal state 0 or the reactivity of the environment is modified, it will jump to a random non-goal state. If the agent is not at the goal state and $x_t^{(2)} = 5$, it can use the actions $0, \ldots, 5$ to jump to any desired state with absolute certainty. As the value of the knob $x_t^{(2)}$ is reduced, the agent starts losing control of where it is going, and it is instead more likely to jump to the goal. Now, the final ingredient, the reward, is:

$$R_t(x_t^{(1)}, x_t^{(2)}) = \begin{cases} 1.0, & \text{if } x_t^{(1)} = 0 \text{ and } x_t^{(2)} = 5 \text{ and } t = T \\ 0.33, & \text{if } x_t^{(1)} = 0 \text{ and } x_t^{(2)} < 5 \text{ and } t = T \\ 0.0, & \text{otherwise.} \end{cases}$$

When we put all these pieces together, we have the following situation: to achieve the maximum reward, we want to keep the the knob $x_t^{(2)}$ at its "maximum reactivity" value of 5. But when we do that, *the action that will take the agent to the goal depends on the state it is located at*. Therefore, the

agent needs to *react* to the location that it is at. And when planning ahead, to properly choose the best first action to take, the agent needs to score the options according to this future ability to react to actions. By turning the knob $x_t^{(2)}$ all the way down to 0, we can jump to the goal by taking any action $0, \ldots, 6$, regardless of where we are, i.e., we do not need any reactivity.

We present this factored MDP, initialized at $x_0^{(1)} = 0, x_0^{(2)} = 5$ to both VBP and SOGBOFA-LC*. Their behavior is very different. VBP is aware that it can jump to the goal state $x_T^{(1)} = 0$ at any time as long as it is not there yet. So it just idles in the other states, keeping the reactivity at the maximum level, until the last time step, in which it jumps to the goal, capturing a reward of 1.0. SOGBOFA-LC*, even though it is replanning at each state, can only evaluate the possible actions by considering a single best sequence of non-reactive actions at a time. Since the reactive plans that VBP prefers are not "visible" to SOGBOFA-LC*, it decides to use the first 5 actions to reduce the reactivity of the environment all the way down to $x_{T-1}^{(2)} = 0$. In that way, it is guaranteed to be able to jump to the goal in the last instant without having to be reactive, even if only capturing a reward 0.33.

This example highlights how SOGBOFA-LC* (and MMAP planning in general) struggle with reactive environments in which the best future actions cannot be known at the current time step. Essentially, the agent is myopic to the fact that it will be able to replan and chooses to be conservative and *reactivity avoidant*, proposing sequences of actions that will be guaranteed to work in any scenario, even if this reduces the obtained reward. We can make the difference between VBP and SOGBOFA-LC*'s performance arbitrarily large simply by increasing the number of states and planning horizon.

MAP inference presents the same non-reactivity problems as MMAP, combined with the lack of integration across multiple paths. On this problem, the average reward achieved by a perfect MAP inference agent is $\sim 0.13$.

### I.3 Experimental Details on ICAPS 2011 International Probabilistic Planning Competition Problems

#### I.3.1 Experimental Settings

Six different inference approaches, MFVI-Bwd, CSVI-Bwd, ARollout, SOGBOFA-LC, VI LP, VBP, and a random agent are evaluated on 6 different domains (Crossing traffic, Elevators, Game of life, Skill teaching, Sysadmin, and Traffic) used in the ICAPS 2011 International Probabilistic Planning Competition (Sanner, 2011). Figure 3 shows the average cumulative reward of all domains. Table 2 shows the corresponding objective and reference for each approach. There are 10 instances for each domain. All instances have a horizon of 40 and a discount factor of 1. The cumulative reward is averaged over 30 simulations for each instance and the plotted error bar shows its standard error of the mean. Given the current state $x_t$, the transition probability $P(x_{t+1}|a_t, x_t)$, and the reward function $R$, each approach infers the best next action $a_t$ at each time step. The reward function for all 6 domains only depend on the state $x$. The cumulative reward is the sum of the rewards of the initial state and the following 39 states.

For all inference approaches, we run with a look ahead horizon of both 4 and 9. We follow the settings in (Wu and Khardon, 2022) where the look ahead horizon is truncated if it extends beyond the remaining time steps. The horizon with the higher average cumulative reward is then selected for each instance. The experiment evaluates over two look ahead horizons due to the observation that for some domain instances a longer horizon may lead to worse estimates. For each simulated run, the same set of random numbers are applied to the simulated environment to reduce noise when comparing different inference approaches. All experiments were run on CPU machines in the cloud.

#### I.3.2 MFVI-Bwd

The backward mean field variational inference approach (MFVI-Bwd) is based on the implementation in (Wu and Khardon, 2022 – url: `https://github.com/Zhennan-Wu/AISPFS`), in which the mean field approximation $q_\phi$ is the product of independent factors. $q_\phi$ is first obtained by maximizing the ELBO of the true posterior under the condition that the maximum accumulated reward is reached using the EM algorithm. The action is then selected based on the marginal $q_\phi(\boldsymbol{a})$. In the experiment, the maximum number of iterations is set to 100 and the convergence threshold is set to 0.1 for the EM algorithm. MFVI-Bwd experiments were ran on a single CPU machine with 32 virtual cores. All 30 simulations were ran in parallel for each task instance.

Table 2: Six different inference approaches used for comparison in the ICAPS 2011 International Probabilistic Planning Competition problems and their corresponding objective and reference.

| Inference Approach | Objective | Reference |
|---|---|---|
| MFVI-Bwd | — | (Wu and Khardon, 2022) |
| CSVI-Bwd | — | (Wu and Khardon, 2022) |
| ARollout | $\tilde{F}^{\text{marginal}^{\text{U}}}_{\lambda=0}$ | (Cui et al., 2015; Cui and Khardon, 2016) |
| SOGBOFA-LC | $\tilde{F}^{\text{MMAP}}_{\lambda=0}$ | (Cui et al., 2019) |
| VI LP | $\hat{F}^{\text{planning}}_{\lambda=0}$ | Ours |
| VBP | $\tilde{F}^{\text{planning}}_{\lambda\approx0}$ | Ours |

### I.3.3 CSVI-Bwd

The backward collapsed variational inference approach (CSVI-Bwd) is proposed in (Wu and Khardon, 2022). It uses collapsed variational inference to effectively marginalizes out variables other than the actions to achieve a tighter ELBO. The authors have shown that this approach out performs MFVI-Bwd. CSVI-Bwd experiments were ran with the same hardware setup as MFVI-Bwd.

### I.3.4 ARollout

The Algebraic Rollout Algorithm (ARollout) introduced in Cui et al., 2015 is equivalent to belief propagation when conditioned on actions as shown in Cui et al., 2018. In our experimental setting which the search depth is fixed with no computation time limit, ARollout is equivalent to the original SOGBOFA (symbolic online gradient based optimization for factored actions) approach introduced in (Cui and Khardon, 2016). While ARollout performs approximate aggregate rollout simulations to evaluate each action, SOGBOFA uses the gradient of the accumulated reward to update actions for exploration and can be advantageous when the action space is large given a computation time constraint. In this experiment, we use the results of forward belief propagation to represent ARollout.

For each time step, the action with the highest estimated accumulated reward is selected. The estimated accumulated reward is calculated by running a forward pass on the factored MDP representing the problem conditioned on the next action. ARollout experiments were ran on CPU machines with 2 virtual cores. All simulations were ran in parallel.

### I.3.5 SOGBOFA-LC

SOGBOFA-LC is based on the Lifted Conformant SOGBOFA implemented in (Cui et al., 2019 – url: `https://github.com/hcui01/SOGBOFA`). SOGBOFA-LC has two differences over the original SOGBOFA. First, it uses a lifted graph that saves computation by identifying same operations. This improvement in computation speed doesn't have an effect in our experiment result since we provide enough computation time for the given maximum search depth. Second, it uses conformant solutions, which the evaluation of the next action is based on a linear rollout plan that best supports the action. This is achieved by calculating the gradient with respect to all actions within the search depth.

In our experiment, the search depth is set to 9 or 4 based on the look ahead horizon. The number of gradient updates is set to 500 following the experimental setting in Wu and Khardon, 2022. The allowed time is set to 50000 per iteration, which is sufficient for 500 updates given the assigned depth. SOGBOFA-LC experiments were ran on CPU machines with 4 virtual cores sequentially.

Note that the RDDL files in the repository (`https://github.com/hcui01/SOGBOFA` or `https://github.com/Zhennan-Wu/AISPFS/tree/master/SOGBOFA`) have different initial states from the original competition for some instances in the elevator and skill teaching domains. These differences were corrected to match the original competition settings in our experiments. Modifications were also made to use the same random numbers in the environment as other experiments and to measure the standard error of the mean instead of the standard deviation.

### I.3.6 VI LP

The Variational Inference Linear Programming (VI LP) approach uses the GLOP solver in Google's OR-Tools (Perron and Furnon, 2024) to solve the linear programming (LP) problem derived from each task instance with the target of maximizing the expected accumulated reward. Constraints on states that are specified in the original RDDL problem is added to the LP problem. Among the six domains only the elevator and crossing traffic domains have such constraints. These constraints specify that the elevator/robot cannot be at different floors/locations at the same time step.

The solver is run for each next action at each time step. The next action that has the highest estimated expected accumulated reward based on the solver is selected. See Section 3.1 for more details of this approach. VI LP experiments were run on CPU machines with 16 virtual cores. All simulations were run in parallel. For each iteration, LP solvers for each next action were run concurrently with multiprocessing.

### I.3.7 VBP

Note that setting a value for $\lambda$ is meaningless if the reward can have arbitrary scaling (and for the case of IPPC in which rewards are additive, the scaling is indeed arbitrary). Therefore, we first normalize all rewards to have a maximum point-to-point variation of 1.0. $\lambda$ is chosen as a reward multiplier on top of that normalization. The closer to 0 that we set $\lambda$ then, the closer we are to the additive limit. Although outside the scope of this work, it is actually possible to set $\lambda$ to exactly zero in VBP (the problem remains non-convex, but the message updates within each time step become an LP problem). However, at least in a straightforward implementation, we observed such message updates to not result in a good optimization of our score function, and often not converging. We attribute this to the degeneracy of the solutions in the limit. Using instead a small value of $\lambda = 0.3$ had a favorable effect on convergence, while remaining close enough to the additive limit. Empirically, we found that most problems were not very sensitive to the choice of $\lambda$.

For each time step, VBP messages are propagated concurrently for a maximum of 150K iterations with 0.1 damping. The $\epsilon$ value is annealed every 300 iterations from a value of 1 to 0.01 based on the formula described in Appendix D. The next action with the highest expected accumulated reward is then selected. See Section 3.1 and Appendix D for more details of this approach. VBP experiments were ran on CPU machines with 2 virtual cores. All simulations were run in parallel.

