# OpenReview forum: "What type of inference is planning?"
_NeurIPS.cc/2024/Conference — NeurIPS 2024 spotlight_

### Official Review · Reviewer_us9u · 2024-06-15

**Soundness:** 4
**Presentation:** 4
**Contribution:** 3
**Rating:** 6
**Confidence:** 4

**Summary:**

This paper studies 3 types of factor graph inference for planning and in particular their relationship to optimal planning. The well known $\exp(\lambda R(x, a, x'))$ factor is used. MAP inference computes the maximum energy configuration over states and actions which corresponds to zero posterior entropy. Marginal inference computes the joint over states and actions. Marginal MAP inference set the action entropy in marginal inference to zero. An interesting connection was made with the dual LP formulation of MDP. The method was further applied to factored state MDP. The relationship between MDP stochasticity and inference approximation to optimal planning was studied in the experiments.

**Strengths:**

**Originality & significance**: The paper is original and the three types of inference have not been studied in prior "planning as inference" literature to my knowledge. The connection between the variational formulation and the dual LP formulation is also interesting. In the discussion on stochastic dynamics, the notion of "reactivity" is also interesting and formalizes intuition from prior work to some extent.

**Quality & clarity**: The paper is well written and the notation and exposition are clear.

**Weaknesses:**

There isn't any salient weaknesses of the paper as far as I can tell. My impression is the results are somewhat "expected" given this type of "planning as inference" has been widely studied in prior work, at least empirically.

One thing that would be of interest is how "reactivity" interacts with online re-planning.

**Questions:**

I don't have any outstanding questions for the authors.

**Limitations:**

This paper does not contain a limitations section.

---

> ### Author Rebuttal · Authors · 2024-08-07
>
> Thank you for your time in reviewing this paper. We would like to address one of your comments:
>
> **One thing that would be of interest is how "reactivity" interacts with online re-planning.**
>
> We do a worked out example in Appendix F.2, in which we show how for some MDPs online replanning is not as good as using a reactive policy when planning. In essence, an agent not only benefits from the ability to replan, but also from knowing ahead of time that it will have the ability to replan.

---

> ### Comment · Reviewer_us9u · 2024-08-08
>
> I thank the authors for the response. The worked example is very helpful.

---

### Official Review · Reviewer_qHG5 · 2024-07-10

**Soundness:** 3
**Presentation:** 3
**Contribution:** 4
**Rating:** 7
**Confidence:** 4

**Summary:**

This paper investigates the concept of "planning as inference", which frames planning problems (i.e. coming up with actions to reach a goal or attain reward) using the vocabulary and mathematics of probabilistic inference. The paper surveys existing formalizations of this broad concept, finding that none of them exactly correspond to the standard planning notion of computing a policy that maximizes expected cumulative reward. In response, the authors introduce a new variational objective, $F_\text{planning}(q)$ the optimization of which is a generalization of the objective of maximizing expected cumulative reward using some (potentially reactive) policy $\pi$ (instead of just finding a single action sequence $\mathbf{a}$ that performs this maximization, as in MMAP). They also show that this objective differs from existing inference objectives (MAP, MMAP, and Marginal inference) in terms of the entropy terms associated with their variational formulations. The authors then derive several practical methods for optimizing $F_\text{planning}(q)$ based on linear programming (VI LP) or loopy belief propagation (VBP) to compute the optimal $q$ (which corresponds to the action policy $\pi$). In both synthetic experiments and benchmark IPC problems, they show that these methods are competitive with other planning-as-inference algorithms based on other inference formulations, and confirm that optimizing $F_\text{planning}(q)$ leads to (desirably) different properties than MAP or MMAP inference, e.g. in terms of policy reactivity.

**Strengths:**

This was a detailed, interesting, and well-substantiated paper that addresses an important conceptual question for the field of planning-as-inference: What inference objective corresponds to the standard decision-theoretic notion of maximizing expected cumulative reward? To my knowledge, this question hasn't been adequately addressed, and this paper does a very thorough job of providing an answer --- one that I believe will be valuable for future practitioners in the field.

Overall, the paper was well-presented for researchers already with an understanding of planning-as-inference, and the theoretical results were mostly well-explained (see later comments for how the exposition could be improved for a broader audience, as well as several theoretical claims that need to be better justified / clarified). I especially appreciated the explanations as to how the optimization of  $F_\text{planning}(q)$ differs from other inference objectives (vis a vis policy reactivity), and the derivation (mostly in the Appendix) of how a variant of their objective can capture determinization in planning (which makes clear the variational gap this introduces).

Regard soundness, the proofs that I checked (in Appendices A, B, ad E) appear to be correct, and the experiments provide sufficient validation for both the new objective and the approximate methods they use to optimize it. Even though these approximate methods mostly match existing methods for planning-as-inference like SOGBOFA-LC, the experiments show that in at least some cases (e.g. higher entropy cases or cases requiring reactivity), optimizing the right objective (i.e.  $F_\text{planning}(q)$) leads to higher cumulative reward.

**Weaknesses:**

In my opinion, the main way this paper could be improved is in exposition, especially for a broader audience than people already familiar with planning-as-inference, factor graphs, and variational inference. The introduction right now is very short, and then paper goes straight into mathematical background. IMO, some of the discussion in the Related Work should be moved into the Introduction itself, and I think the Related Work probably is best placed after the Introduction. I also think both the Abstract and the Introduction could foreground the intuitive / standard notion of planning as "maximizing expected cumulative reward", and communicate that their variational formulation of planning-as-inference both reproduces this objective and generalizes it, whereas other inference objectives do not. This, to me, is one of the main upshots of the paper, and it should be communicated clearly from to get-go as a motivation. Right now, the idea that maximizing $F_\text{planning}(q)$ generalizes "maximize expected cumulative reward" only starts appearing in Table 1, and  Section 3.1.

As an additional organizational suggestion, after introducing the Background, I think it is probably better to cover the content in Section 4 first, by introducing $F_\text{planning}(q)$ as a variational objective, then comparing it to the other existing inference objectives. After you have established that $F_\text{planning}(q)$ is the "right" objective for planning, then you can go into the content of Section 3.2 as a separate section that is focused on practical methods for optimizing the planning objective. By organizing things in this way, you first clearly motivate the desirability of $F_\text{planning}(q)$ as an objective, which then motivates practical methods for optimizing $F_\text{planning}(q)$.

Aside from exposition, I think a number of theoretical claims that the paper makes could be better explained and clarified. In particular, it was not obvious to me that the ranking of inference types in Section 4.1 could be read off from the entropy terms in Table 1 -- there should be an associated proof in either the main paper or Appendix. There are also a few notational choices that I think need to be clearly defined before use, and several sentence discussing the relationship of  $F_\text{planning}(q)$ to other inference objectives that could be clarified. I will detail these more in the Questions section.

I think the experiments could be made even stronger by comparing against e.g. Monte Carlo methods for planning-as-inference (e.g. Piche et al. (2018)) and model-based planning algorithms for MDPs (standard value iteration, RTDP, etc.), but this is not strictly necessary.

**Questions:**

Line 62: "For a general factor graph f(x, a)" -- please define what f(x, a) is. Even among probabilistic inference practitioners, the factor graph representation of a inference problem may not be widely known (compared to e.g. computing or sampling from a posterior distribution).

Line 62: I think the bra-ket notation for expectations should probably be defined somewhere, since it is less commonly used than the expectation operator. And perhaps the paper should consistently use one notation or the other.

Table 1: Could you provide derivations or explanations of the variational identities for Marginal, MMAP, and MAP somewhere (e.g. the Appendix), just as you've done for $F_\text{planning}(q)$? Right now it's not obvious to me why the conditional entropies end up differing in the way shown in column 2, and it seems like there should be some fairly straightforward intuition as to why, e.g. MMAP requires subtracting out $H_q(a_t)$, and why Marginal has $H_q(x_{t+1}, a_t | x_t)$ instead of $H_q(x_{t+1} | a_t, x_t)$ as the conditional entropy term.

Table 1 and Eq 1: $R(\mathbf{x}, \mathbf{a})$ should be explicitly defined some where as the cumulative reward.

Line 159: Some derivation of the computational complexity would be good (e.g. in the Appendix).

Lines 180-181: "none of which corresponds with the exact “planning inference” from this work, which is exact." --- exact with respect to what? Exactly captures the notion of maximizing expected cumulative reward? Exact in the sense that $F_\text{planning}$ is not a lower bound of itself? I think this should be clarified.

Line 182: "First, from inspection of the entropy terms it is trivial to see that.." --- to me this wasn't trivial for all the comparisons shown, and there should be some explanation and derivation. In particular, it was not obvious to me why the entropy term for Marginal$^\text{U}$ should be lower the entropy term for MAP, which is zero.

Lines 185-186: "Since the tightness of a lower bound is an indication of its quality, it follows that MMAP inference is no worse and potentially better than all other common types of inference." --- I think it is confusing for this sentence to follow right after Equation (8), because it suggests that the "best" objective is $F_\text{marginal}$, even though the whole point of the paper is that  $F_\text{marginal}$ does not capture our standard notion of planning. I think it should be clarified that the MMAP objective is a lower bound to the planning objective in particular, which is the true objective the paper cares about. You could do this with a sentence like "Except for $F_\text{marginal}$, which is not a planning objective in the sense of maximizing expected (exponential) utility,  $F_\text{MMAP}$ is the best lower bound to  $F_\text{planning}$".

**Limitations:**

While there is no explicit limitations section, the authors adequately discuss the approximate nature of their proposed optimization methods, and also discuss cases where their proposed objective is equivalent (and hence does not improve on) existing planning-as-inference formulations (namely, cases without deterministic dynamics).

There is some discussion of this already, but I think it would be good for the paper to better recognize when Marginal$^\text{U}$ might be desirable as an objective, as this corresponds to a notion of "soft planning" that is similar to Boltzmann-rational models of action selection and max-entropy inverse RL (Ziebart et al, 2010). Sometimes, our goal is not to derive a reward-optimal policy, but to get a policy that is stochastic (because we desire diversity, or because we want to model human planning).

The paper is largely theoretical in nature, and discussions of societal impact are not immediately relevant.

---

> ### Author Rebuttal · Authors · 2024-08-07
>
> Thank you for your careful review this paper, and for your insightful comments and suggested improvements, which we will incorporate.
>
> **Please define what f(x, a) is**
>
> We tacitly take f(x, a) to correspond to the factor graph of Fig. 1 [left], but this is not explicitly mentioned in the text. We will define fully.
>
> **Define the bracket notation or convert everything to expectations for consistency**
>
> We will convert everything to expectation notation.
>
> **Could you provide derivations or explanations of the variational identities for Marginal, MMAP, and MAP somewhere (e.g. the Appendix), just as you've done for $F_\text{planning}(q)$?**
>
> All the other variational expressions have been provided somewhere in existing literature, so we didn't rederive them in this work. But we can indeed include them in an appendix if the reviewer feels that it would improve clarity.
>
> Marginal inference: This is the standard VI problem, see e.g., (Jordan et al., 1999).
>
> MMAP inference: This result is derived in (Liu and Ihler, 2013).
>
> MAP inference: Taking the variational problem of marginal inference and setting the entropy to zero (sometimes by multiplying it with a temperature that goes to 0) results in an LP problem that corresponds exactly to maximum a posteriori inference (Weiss et al., 2012). Further relaxing the marginal polytope into a local polytope gives rise to most well-known methods for approximate MAP inference, such as dual decomposition. See e.g., (Sontag et al., 2011).
>
> **$R({\pmb x}, {\pmb a})$ should be explicitly defined some where as the cumulative reward**
>
> Oftentimes (and indeed in our experiments), rewards do not depend on the action and the above expression can be simplified to $R({\pmb x})$. This is already defined in line 48. We will define the more general case $R({\pmb x}, {\pmb a})$.
>
> **Some derivation of the computational complexity would be good (e.g. in the Appendix).**
>
> The computational complexity is mentioned in line 159, we will add a derivation to the appendix in the final version.
>
> **"none of which corresponds with the exact “planning inference” from this work, which is exact." --- exact with respect to what?**
>
> The quantity of interest of "planning inference" is the expected utility of an optimal planner. When running "planning inference" in a standard MDP, the result is exactly the best expected utility. This is exact with respect to, for instance, value iteration.
>
> This means that, in a standard MDP, an agent that follows the best action as prescribed by "planning as inference" is an optimally-acting agent that will attain maximum reward, regardless of the MDP. However, given an agent that follows the optimal action according to any other type of inference (even if such inference is exact), can be fooled by a specially crafted MDP, and in general will not perform as well as the "planning inference" agent. We give a worked out example of this in appendix F.2, in which an MMAP agent is led astray by an MDP that is crafted to need high reactivity. Also, Figure 1[right] illustrates this point, where all inference types have a negative advantage, and $F^\text{planning}$ has exactly 0 advantage.
>
> **First, from inspection of the entropy terms it is trivial to see that.." --- to me this wasn't trivial for all the comparisons shown (...) In particular, for Marginal$^\text{U}$**
>
> Yes, an appendix with a proof of all the sequential bounding will be helpful, we will add it in the final version.
>
> In particular, marginal inference can be combined with different normalizing constants. Regardless of the normalizing constant, its behavior is the same and the resulting planner will take the same actions and achieve the same reward. But in terms of bounding, different normalizing constants will have different effects.
>
> Your comment made us realize that a more in-depth analysis of the possible normalizing constants and their effect on bounding should be included in the paper. We will do that in the final version.
>
> For upper bounding, the marginal entropy term
>
> $$H^\text{Marginal}({\pmb q}) = H_{\pmb q}(x_1) + \sum_{t=1}^{T-1} H_{\pmb q}(x_{t+1}, a_t| x_t)$$
>
> clearly upper bounds the "planning inference entropy term"
>
> $$H^\text{Planning}({\pmb q}) = H_{\pmb q}(x_1) + \sum_{t=1}^{T-1} H_{\pmb q}(x_{t+1}|x_t, a_t)$$
>
> since $H_{\pmb q}(x_{t+1}, a_t| x_t) = H_{\pmb q}(x_{t+1}|  a_t, x_t)+ H_{\pmb q}(a_t| x_t)$ and entropies are non-negative.
>
> For lower-bounding, we can use a uniform weighting over trajectories and use the entropy term
> $H^{\text{Marginal1}^\text{U}}({\pmb q})
> = H_{\pmb q}(x_1) + \sum_{t=1}^{T-1} H_{\pmb q}(x_{t+1}, a_t| x_t) - (T-1)\log N_aN_s^{N_e} - \log N_s$
>
> It is clear that $H_{\pmb q}(x_1) \leq \log N_s$ and $H_{\pmb q}(x_{t+1}, a_t| x_t) \leq \log N_aN_s^{N_e}$, with equality being achieved when $\pmb q$ is a uniform distribution. Therefore, $H^{\text{Marginal1}^\text{U}}({\pmb q})$ is always non-positive, as required to lower bound $H^\text{MAP}({\pmb q})$.
>
> If instead, we use a uniform weighting over actions, as suggested in our paper we have
> $H^{\text{Marginal2}^\text{U}}({\pmb q})
> = H_{\pmb q}(x_1) + \sum_{t=1}^{T-1} H_{\pmb q}(x_{t+1}, a_t| x_t) - (T-1)\log N_a$
>
> Since $H_{\pmb q}(a_t)\leq \log N_a$, it follows that $H^{\text{Marginal}^\text{U}}({\pmb q}) \leq H^\text{MMAP}({\pmb q})$
>
> where
>
> $H^\text{MMAP}({\pmb q}) = H_{\pmb q}(x_1) + \sum_{t=1}^{T-1} H_{\pmb q}(x_{t+1}, a_t| x_t)-H_{\pmb q}(a_t)$
>
> If we further assume deterministic dynamics, then $H_{\pmb q}(x_1) = 0$ and $H_{\pmb q}(x_{t+1}, a_t| x_t)\leq \log N_a$, thus making $H^{\text{Marginal2}^\text{U}}({\pmb q})$ non-positive, as required to lower bound $H^\text{MAP}({\pmb q})$.
>
> **I think it should be clarified that the MMAP objective is a lower bound to the planning objective (...) with a sentence like (...)**
>
> Your phrasing is clearer, we will adapt it and adopt it.

---

> > ### Comment · Reviewer_qHG5 · 2024-08-12
> > **Thank you for the response.**
> >
> > Thank you for the response, and for the explanation of the sequential bounding in particular, which clarified my confusions. Do include these derivations in the Appendix.
> >
> > Regarding derivations of the variational expressions, I think it's up to the authors to include the full derivations in the Appendix, but I do think readers will probably find it helpful if some explanation or intuition is provided. For Marginal$^U$ inference, for example, I can see that the $H(q)$ term is really just some kind of KL-divergence between $q(x, a)$ and the (uniform) prior over trajectories or actions. For MAP, I can also see the intuition for why $H(q)$ is zero, because in MAP inference you are just interested in finding a point estimate, and so $q$ should have zero uncertainty (and hence zero entropy). For MMAP, it was helpful for me to realize the entropy term is just equal to $H_q(x_{1:t}, a_{1:t}) - H_q(a_{1:t}) = H_q(x_{1:t} | a_{1:t})$, which intuitively makes sense because in MMAP, you only want to marginalize over $x_{1:t}$ for a fixed value of $a_{1:t}$, and hence only care about the randomness in $x_{1:t}$ conditional on $a_{1:t}$. I think these sorts of intuitive explanations would be helpful to include somewhere.
> >
> > I'll be keeping my score of 7. I encourage the authors to restructure the exposition of their paper in some of the ways I suggested, so that the paper will be more accessible to others and have a greater impact.

---

### Official Review · Reviewer_vctT · 2024-07-12

**Soundness:** 3
**Presentation:** 2
**Contribution:** 3
**Rating:** 6
**Confidence:** 3

**Summary:**

This work introduces a variational inference (VI) framework to characterise different types of inference for planning, specifically for finite-horizon MDPs represented as probabilistic graphical models. They also develop an inference algorithm (VBP) that takes inspiration from the loopy belief propagation and adapts it to planning. An experimental evaluation seems to confirm the theoretical analysis.

**Strengths:**

The idea is very interesting and original. They also performed an extensive experimental analysis.

**Weaknesses:**

The paper has two major flaws. First, the text is poorly written, and it definitely doesn’t meet the standards of a high profile conference as NeurIPS. In particular,

1) The introduction and the first subsection of the backgrounds have zero references.

2) The paper is not self-contained. I understand the limit of space, but the background section doesn’t provide all the necessary background. Moreover, the experiment on “Reactivity avoidance” is interesting, but most of the explanation is in the appendix, making it hard to grasp with the sole information in the paper.
3) The notation is not always properly introduced, for example:

 - At the beginning of 2.2 I would explicitly write the equations for the different types of inference. Considering how central they are to the paper, and the fact that the work if theoretical, I would expect a more formal definition.

- The risk parameter \lambda is introduced in section 3 without any context on what that means in terms of planning. It is not in the standard formulation of MDPs, nor introduced in your background section.

- The “discussion” is not really a discussion, maybe we can call it a conclusion, but anyway it looks rushed and not very “conclusive”
See detailed comments below for more examples.

Second, the soundness of the contribution is undermined by missing details and not very convincing results. In particular:

4) Your definition of “planning inference”, that is central to the paper, is too fuzzy. On line 96 you state that your definition of “planning inference” corresponds to the entropy in Eq.4, however later on line 181, you state that your definition of inference is exact. This I think is a bit confusing. VI is by nature an approximation. I presume I understand what you want to say, nevertheless I think it should be clarified.

5) The discussion on the different types of inference lacks rigour. The notation is not always clear, and most importantly, it is not clear what is the impact of this finding on the planning community. What does it mean in practice that “planning inference” is different from all the others in stochastic settings?

6) The experiments, despite being extensive, don’t show a significant impact coming from the new “planning inference” introduced by the authors. See my questions to the authors for more details.

**Questions:**

1) Why do you focus on finite-horizon MDPs? Do your result easily translate to infinite-horizon MDPs?
2) In the introduction (line 27) you mention value iteration, but then any connection with classic MDP algorithms (and Bellman update) is missing. Moreover, you focus on “best exponential utility”, why not the classic “maximum expected utility”? Can you elaborate on this? I think there is a sort-of hint on this in section 3.3 but it is not exactly clear to me.
3) In Figure 2 (right) the x-axis should be in [0,1] if you are plotting the normalized entropy, why is it not the case?
4) Connected to the previous question, how do you explain the fact that the different methods converge again when the stochasticity increases? From 4.2 you seem to suggest that the more stochastic the environment, the more your solution should be preferable. Am I missing something?
5) In Figure 3, it looks like SOGBOFA-LC performs slightly worse on game of life than sysadmin. Doesn’t it go against the intuition that MMAP should degrade when the stochasticity increases?
6) On line 276 you say that you notice a *significant* advantage of your proposal wrt to SOGBOFA-LC. I’m not sure if I agree with it. Only for the game of life there is clear improvement, but in that case a Rollout is performing better. Can you elaborate on this?



Minor comments:

- In the abstract, the sentence “...show that all commonly used types of inference correspond to different weightings of the entropy terms” is a bit misleading since these are already known results, as you point out in section 2.2. I recommend re-writing in order to not set the wrong expectations to the reader.

- Some notation is not properly introduced: line 62 (angular brackets, I assume it’s a shorthand for the expected value that might be standard in VI but not in planning communities), line 82 (the limit), line 158 (n_a, n_s), etc.

- Line 44, “small subset of x_t” is an assumption. In principle, x_t^(i) can depend on all x_t-1

- Line 64, I think the “-” before the log shouldn’t be there

- Better avoid contractions and other “informal abbreviations” e.g. “let’s” (line 50) or “wrt” (line 176)

- Line 289, “further the understanding”, I think you want to remove that “the”

- a few more words on the “planning as learning” on line 210, and how that is different from your analysis could be interesting.

- There is a lot of math (that I appreciate) but some intuitive explanation could very much improve the presentation of the results.

**Limitations:**

Okay.

---

> ### Author Rebuttal · Authors · 2024-08-07
>
> Thank you for your time reading this paper, we hope that you will find the following clarifications useful in its judgement.
>
> **Why do you focus on finite-horizon MDPs? Do your result easily translate to infinite-horizon MDPs?**
>
> When thinking of planning as inference in a factor graph, it is simpler to consider a finite factor graph that has been unrolled for a finite number of steps $T$. Classical references such as (Levine, 2018) do the same.
>
> Practically speaking, when solving an intractable MDP (such as a factored MDP with an exponentially large state space at each time step), rewards beyond a certain horizon rarely affect the agent. As an example, competitors in the IPPC were given a finite MDP of horizon 40, but they often designed agents that considered and even shorter horizon. Even if an MDP is infinite-horizon, an approximate finite-horizon agent will probably have similar performance to an approximate infinite-horizon agent, since the approximations have a stronger effect than considering the horizon finite vs. infinite.
>
> Theoretically speaking, it is possible to extend our work to deal with infinite horizon MDPs by using the regular structure of the graph. In an infinite MDP, the forward and backward messages at all timesteps should be identical (discounting edge effects), so this becomes a highly symmetric inference problem. Lifted BP was designed for efficient inferences in highly symmetric models. The recipe that we followed to extend standard loopy BP to planning should apply to extend lifted BP to planning.
>
> **In the introduction (line 27) you mention value iteration, but then any connection with classic MDP algorithms (and Bellman update) is missing**
>
> The connection is mentioned in Section 3.3, where the backward updates correspond exactly with Bellman updates. Note that the definition of $Q(x_t,a_t)$ (in line 163) happens to coincide exactly with the Q table in reinforcement learning. This, together with the backward updates corresponds to value iteration.
>
>
> **You focus on “best exponential utility”, why not the classic “maximum expected utility”**
>
> The best exponential utility is identical to the“maximum expected utility” in the limit $\lambda \rightarrow 0$. Therefore, this is a strict generalization for all $\lambda$.
> Section 3 has three references (Marthe et al., 2023; Föllmer and Schied, 2011; Shen et al., 2014) that explain the role of $\lambda$ in planning. Essentially, it is a parameter controlling how risk-seeking an agent is. Using a generic $\lambda$ allows us to formulate the utility in Eq. (1) in which the exponentiated reward and the dynamics interact multiplicatively. This is critical to express them as a factor graph, since in factor graphs all factors interact multiplicatively.
>
> **In Figure 2 (right) the x-axis should be in [0,1] if you are plotting the normalized entropy, why is it not the case?**
>
> Good catch, we had normalized the entropy in the x-axis for the [left] plot but not for the [right] one. We have corrected this. Nothing changes other than a different labeling in the x-axis, which now runs from 0 to 1.
>
> **how do you explain the fact that the different methods converge again when the stochasticity increases?**
>
> This is an expected result. When the dynamics are moderately stochastic, a single sequence of actions (as in MMAP) becomes more inadequate to solve the planning problem, and we need more reactivity to the environment to plan. Thus, as stochasticity grows, our proposed method becomes increasingly better than, e.g., MMAP. However, if the stochasticity grows too much, the agent starts to lose any ability to control the environment and this advantage is lost. Think of the extreme case: When dynamics are fully stochastic, the agent follows a random walk regardless of which action is taken. In that extreme regime, any planning method will have the same performance: that of a random walk.
>
> **In Figure 3, it looks like SOGBOFA-LC performs slightly worse on game of life than sysadmin. Doesn’t it go against the intuition that MMAP should degrade when the stochasticity increases?**
>
> That would have been our expectation as well. However, the stochasticity of game of life and sysadmin are very similar (17% and 23%, respectively), so other differences can have a stronger effect. For instance, the structure of the MDP describing the game of life depends, on average, on more entities from the previous time slice, as compared with sysadmin. It might also require more reactivity for succesful control. These differences make the gap between VBP and SOGBOFA-LC wider than expected when judging purely from a stochasticity level perspective.
>
> **you say that you notice a significant advantage of your proposal wrt to SOGBOFA-LC. I’m not sure if I agree with it.**
>
> Yes, this requires more qualification. The advantage is significant in the statistical sense for most instances of game of life and sysadmin (note the lack of overlap of the uncertainty regions in most instances). For the other problems, which are mostly deterministic, it is hard to say whether VBP or MMAP is performing better across the board, as expected since they are very deterministic. ARollout on the other hand is a bit hit or miss, since it integrates over all actions rather than optimizing over them. It works well for game of life, but for the very deterministic elevators in which one can derive a clear plan of action, ARollout performs very poorly.

---

> > ### Comment · Reviewer_vctT · 2024-08-11
> >
> > I thank the authors for their detailed rebuttal and will increase my score to a weak accept.
> > I hope the authors can take some of the criticism into account when preparing the camera ready version.

---

> ### Author Response · Authors · 2024-08-07
>
> **The introduction and the first subsection of the backgrounds have zero references.**
>
> This work is related to a vast amount of literature from classical planning, reinforcement learning, variational inference, influence diagrams, etc. While it is impossible to cite all the related work, we have included a selection in our Section 5, "Related work", trying to at least touch all relevant subfields.
> We already have on our radar some additional citations that we want to include in the final version and are happy to hear suggestions for any glaring missing citations.
>
> The reason why all citations from the introduction have been moved Section 5 is that we wanted to be able to connect each citation with a type of inference, and potentially discuss other details about it in relation with our VI framework, and this was only possible after developing the theory sections of this paper.
>
> We will add MDP references to Section 2.1.
>
> **The paper is not self-contained. I understand the limit of space, but the background section doesn’t provide all the necessary background.**
>
> As mentioned before, this work connects multiple fields, so detailed background about all of them cannot be included. Some degree of familiarity with factor graphs, variational inference, loopy BP, and reinforcement learning does indeed help with the reading. We have included the minimal background required to be able to derive our results from either first principles or cited sources. While we are unlikely to be able to fit much more information in the background section, we would like to know if the reviewer has a particular result in mind that could be cited or included to make this work more self-contained.
>
> **The notation is not always properly introduced, for example:**
>
> **(a) Explicitly write the equations for the different types of inference (...) I would expect a more formal definition**
>
> The explicit expression for each type of inference appears in the second column of Table 1. This is a precise definition of each type of inference on the factor graph of Figure 1[Left].
>
> **(b) The risk parameter $\lambda$ is introduced in section 3 without any context on what that means in terms of planning. It is not in the standard formulation of MDPs, nor introduced in your background section.**
>
> Section 3 has three references (Marthe et al., 2023; Föllmer and Schied, 2011; Shen et al., 2014) that explain the role of $\lambda$ in planning. It controls how risk-seeking the agent is. Although this is not standard in MDPs or RL, it is not a concept that we introduce in this paper, but rather, a concept that has already been developed for planning in MDPs and that we leverage to include both the dynamics and the rewards within the same factor graph, since all the terms become multiplicative in this formulation. This is a generalization of the standard utility in MDPs, so we do not lose generality.
>
> **(c) Your definition of “planning inference”, that is central to the paper, is too fuzzy**
>
> Theorem 1 defines the "planning inference" objective, which is Eq. (2), whose terms are further defined in Eqs. (3) and (4). Approximations are introduced in Section 3.2 to make this objective tractable for factored MDPs.
>
> **You state that your definition of inference is exact. This I think is a bit confusing. VI is by nature an approximation.**
>
> To clarify: VI is exact when the variational distribution is arbitrarily flexible and is optimized completely. In other words, the evidence lower bound (ELBO) matches the exact evidence if (a) the variational distribution can fit the posterior arbitrarily well and (b) the optimization finds the maximum of the ELBO. VI is thought of as approximate because typically none of these two conditions are met. In the case of a standard tractable MDP, both conditions are met for "planning inference", and it produces the exact result. This makes sense, since other techniques (such as value iteration) also allow for exact planning. In the case of an intractable factored MDP, we need to introduce approximations, and the proposed VBP no longer corresponds to exact planning.

---

### Official Review · Reviewer_ppF9 · 2024-07-13

**Soundness:** 4
**Presentation:** 3
**Contribution:** 4
**Rating:** 8
**Confidence:** 5

**Summary:**

This paper shows that planning in an MDP can be posed as a specific form of
inference in a graphical model; in this context of inference, different forms
of inference can be applied, with different results. Additionally, alternate
forms of inference such as variational techniques can be used, which allow
better plans to be inferred than existing baselines.

This is not a perfect paper, but overall it is a good idea, well executed and
I recommend it for acceptance.

**Strengths:**

- The primary strength of this paper is the technical contribution, which is
how the paper places the planning problem in the context of inference in
graphical models. The relationship between planning and inference is not
novel, but the generalisation of this relationship to different forms of
inference is novel.

- The paper (mostly) shows how the policy for a standard, flat MDP can be
found by solving for the variational posterior over states, and also shows how
a variant of belief propagation can be used to find this posterior.

- The paper generalises this result to factored MDPs, and shows how belief
propagation is even more useful.

- The paper evaluates the VBP technique against gradient descent baselines and
  exact solution techniques on both 5000 synthetic MDPs and also 6 domains
  from the internation planning competition. The paper shows that VBP is very
  close to the exact solution and outperforms the other baselines on the
  synthetic domains. On the IPC domains, the VBP seems to match the
  performance of the best baseline across all IPC domains (different baselines
  have different performance across different domains -- VBP seems to match
  the best one on each domain).

**Weaknesses:**

The paper has two primary weaknesses:

- There is no discussions of the weaknesses of the inference-based
  technique. It is not clear if the VBP is computationally more costly than,
  for instance, the gradient-based SOGBOFA techniques. Timing information
  would have been very helpful.

  It is also the case that while VBP consistently matches the best performing
  baseline, it does not seem to outperform any of the baselines. It would have
  been helpful to know why VBP isn't able to match the exact planning
  technique -- where is the loss in performance coming from?

- Some of the writing is less than clear. For instance, the exponential
  utility is introduced in table 1, and is justified on page 3 with a short
  reference to being "more suitable for a factor graph representation" but
  more explanation would have been helpful. Please note in the author response
  that the concern is not that the exponential utility is problematic, but the
  text needs to spend more time addressing and justifying this.

  Similarly, the introduction of the $\lambda$ risk setting is not the common
  case, and needs more justification. It's not clear that the $\lambda$ term
  adds much to the primary point of the paper, and seems to be set to 0 for
  most (if not all) of the experimental results. Furthermore, appendix F.3.7
  indicates that $\lambda$ has no meaning if "if the reward can have arbitrary
  scaling". This is a crucial point that should be in the main body of the
  paper, and in fact the paper might be clearer without incorporating
  $\lambda$ at all since it does not add much to the explanation of planning
  as inference.

**Questions:**

- Are there any computational constraints on the VBP? Is it comparable in
  speed to the gradient descent techniques?

- Are there convergence issues? The paper briefly discusses this but more
  detail is needed.

- Can the other forms of inference (e.g., $F^{marginal}$, etc.) map exactly to
  different forms of planning? The experimental results seem to suggest this,
  e.g., $F^{marginal}$ maps to ARollout, but it would be helpful to know this
  earlier in the paper.

**Limitations:**

The authors did not include a discussion of limitations, and this is one of the
weaknesses of the paper.

---

> ### Author Rebuttal · Authors · 2024-08-07
>
> Thank you for your careful reading and positive impression, we will take your comments into account to improve the clarity of the final manuscript.
>
> **Are there any computational constraints on the VBP? Is it comparable in speed to the gradient descent techniques?**
>
> VBP computational and convergence behavior mirrors that of standard loopy belief propagation.
>
> Per iteration, the computational cost is of the same order as gradient-based SOGBOFA techniques. However, unlike SOGBOFA, it is not finding a single sequence of actions, but what in effect is an approximate policy. This is more involved an often requires more iterations, particularly in very loopy graphs. In the IPPC problems, VBP was slower than SOGBOFA. VBP does not have a theoretical advantage over SOGBOFA when dynamics are very deterministic, and it is likely to be slower, so in those cases SOGBOFA might be a better alternative.
>
> **Are there convergence issues? The paper briefly discusses this but more detail is needed.**
>
> Just like when using loopy BP for inference, there are no convergence guarantees, and the presence of convergence actually correlates with good performance. There are many methods in the literature that address the non-convergence of loopy BP, turning it into a convergent algorithm (often using a double-loop algorithm, or using a concave approximation to the entropy terms to make the problem monomodal). All of these methods can be applied to VBP. Interestingly, although all of these approaches result in convergent algorithms, the performance doesn't significantly improve. This was our observation as well. Our convergent variant VI LP did not typically perform as well or as robustly as VBP, despite being convergent.
>
> **Can the other forms of inference (e.g., $F^\text{Marginal}$
> , etc.) map exactly to different forms of planning? The experimental results seem to suggest this, e.g., $F^\text{Marginal}$ maps to ARollout, but it would be helpful to know this earlier in the paper.**
>
> Yes, different forms of inference map to different forms of planning, we discuss this in section 5 (Related work), but instead of using the $F$ notation, we describe it verbally, for instance pointing out that SOGBOFA corresponds to marginal MAP (which would be $F^\text{MMAP}$). We will further clarify this.
>
>
> **Use of the exponential utility and $\lambda$**
>
> The exponential utility is what allows us to write a compact factor graph that contains both rewards and dynamics. It contains the standard accumulated reward as a special case (when $\lambda \rightarrow 0$), so it is strictly more general. Note that in the standard formulation, rewards are additive, so we cannot include them in the factor graph directly. There is a path to deriving a variational formulation for the standard accumulated reward by using additional auxiliary variables and another step of lower bounding, but it would be an additional complication, and we believe that the introduction of $\lambda$ makes the model more flexible in practice (see below).
>
> This is a pervasive problem when trying to connect planning with inference. See for instance (Levine, 2018). Their Eq. 8 tacitly uses an exponential utility with $\lambda = 1$. The motivation is the same as ours, being able to formulate the dynamics and rewards in the same factor graph. The introduction of $\lambda$ allows us to keep the simple factor graph formulation, while still being able to handle additive rewards. In practice, $\lambda$ would be a free tunable parameter that simply acknowledges the relevance of the scale of the reward when making rewards interact in a multiplicative way.
>
> The IPPC setup corresponds to $\lambda \rightarrow 0$. However, in our internal experiments, we observed that for some problems, other values of $\lambda$ resulted in agents that were able to capture more rewards (even when measured in terms of the standard accumulated reward). The limitations of approximate inference sometimes make agents more conservative, and larger values of $\lambda$ encourage them to take more risks, which allows them to effectively capture more reward, regardless of how you measure it.
>
> A more thorough discussion of the motivation and implications of the use of $\lambda$ parameter will be included.

---

> > ### Comment · Reviewer_ppF9 · 2024-08-09
> >
> > I thank the authors for their clear rebuttal. I have no further questions.

---

### Official Review · Reviewer_QvAf · 2024-07-16

**Soundness:** 3
**Presentation:** 2
**Contribution:** 3
**Rating:** 6
**Confidence:** 2

**Summary:**

The paper proposes a framework to understand, as the title suggests, what type of inference is (probabilistic) planning. The authors make use of variational inference which encompasses marginal, MAP, and marginal MAP (MMAP) inference to theoretically compare the power of such types of inference in bounding the optimal expected reward for probabilistic planning. Furthermore, the authors propose an approximation of planning inference for factored MDPs as they search space is exponential in the input problem. Experiments are run to support the theoretical results and claims.

**Strengths:**

1. The paper poses an interesting theoretical study from a statistical viewpoint on optimisation functions for planning.
2. The main story and results are well presented and easy to understand.
3. The experiments are diverse and well motivated, with results supporting the theoretical claims.
4. The paper provides a good coverage of related work.

**Weaknesses:**

1. The main weakness of the paper is its clarity. The paper can be better self-contained if some of the other sections can be better motivated and/or explained by an additional sentence or two, rather than references to the appendix. Furthermore, some of the equations and technical details seem to be ambiguous and not well defined. See suggestions/comments below.

**Questions:**

1. The introduction mentions that planning inference is the same as `value iteration', but nowhere else in the paper is value iteration nor Bellman backups explicitly mentioned. Is this a typo with variational inference (both have the same acronym), or is this intended? If this is intended, please expand on this relationship. The closest looking result is in Sec. 3.3.
2. Can you explain more what is the meaning and/or impact of the result that $F^{\text{planning}}_{\lambda} \leq F^{\text{marginal}}_{\lambda}$? e.g. Is the marginal inference an overapproximation?
3. What are the terms in the denominator of $H_{\text{MDP}}$ and how is it computed for the IPPC domains?

Suggestions/comments
- It would be helpful for the reader to provide definitions of some notation in the variational inference section. More specifically it would be helpful to define what is a factor graph and the function $\langle \cdot \rangle_{q}$ in the `energy term'.
- It could be worth mentioning what is the type of $\mathbf{a}$ and $\mathbf{x}$ in Eqn. 1. I assume now that it is a list of $a$s and $x$s of size $T$
- line 114: never access to the full joint -> never access the full joint
- The motivation and explaination for the Bethe approximation can be brought earlier rather than referring to the appendix.
- The term $H_{\text{Bethe}}$ is not used in the equation under line 118. I only see $H_{\text{Bethe}}^{\text{planning}}$ and $H_{\text{Bethe}}^{\text{marginal}}$
- Sec 3.3. forward updates: the sum should not range over $a_t$ if it is defined as the argmax of $Q(x_t, a_t')$. Furthermore, either bring the equality out of the equations, or remove the $a_t=$s in the forward updates and posterior equations.
- The meaning of $H_{\text{MDP}}$ as normalised entropies can be explained with an additional sentence rather than referring to the appendix. Furthermore, even when looking at the appendix, none of the $N_e, N_a, N_s$ terms seem to be defined anywhere in the paper. Also is this a new metric or something that exists? There does not seem to be a reference associated with it.

**Limitations:**

The authors correctly fill in the checklist and justify their answers.

---

> ### Author Rebuttal · Authors · 2024-08-07
>
> Thank you for your careful reading and catching several typos. We have corrected the paper accordingly and include further explanations below.
>
> **The introduction mentions that planning inference is the same as `value iteration', but nowhere else in the paper is value iteration nor Bellman backups explicitly mentioned. If this is intended, please expand on this relationship.**
>
> We will explicitly mention the connection with value iteration and Bellman backups. Planning inference is exact in standard (non-factored) MDPs, and corresponds exactly to value iteration. The backward updates of Section 3.3 together with the definition of $Q(x_t, a_t)$ in line 163 can be recognized as the standard Bellman backups, although written with slightly different notation, since these are messages. In particular $Q(x_t, a_t)$ is exactly the Q-function.
>
>
>
> **Can you explain more what is the meaning and/or impact of the result that $F_\lambda^\text{planning}\leq F^{\text{marginal}}_{\lambda}$? e.g. Is the marginal inference an overapproximation?**
>
> $F^{\text{marginal}}_{\lambda}$ corresponds to running marginal inference on the _unnormalized_ factor graph from Figure 1, the same on which we run planning inference. That factor graph is missing a prior term for the actions. This means that rather than _averaging_ over all the possible sequences of actions, we are _summing_ over all the possible sequences of actions. In the deterministic case it is obvious that this results in an overapproximation, since we are summing the "correct" total reward corresponding to the optimal sequence of actions plus the rewards obtained by all other sequences of actions. This intuition, although not as direct, carries over to the stochastic case, so yes, it is an overapproximation in the general case.
>
> **What are the terms in the denominator of $H_\text{MDP}$ and how is it computed for the IPPC domains?**
>
> A factored MDP such as the one in Figure 1[Right] describes, among other things, the dynamics of multiple _entities_ that are locally independent. E.g., the distribution of $x^{(1)}$ is independent of the distribution of $x^{(2)}$ given the action and values of the entities in the previous time slice. We can separately compute the conditional entropies $H(p(x_{t+1}^{(i)}|x_t,a_t))$ for each entity $i$, conditional on each possible value $x_t, a_t$ of the previous time slice.
>
> The normalized entropy is defined as
> $$
> H_\text{MDP} = \frac{\sum_{i,x_t,a_t} H(p(x_{t+1}^{(i)}|x_t,a_t))}{N_eN_aN_s^{N_e}\log N_s}
> $$
> (note that there's a typo in the manuscript). The largest value of the conditional entropy $H(p(x_{t+1}^{(i)}|x_t,a_t))$ is achieved when dynamics are purely random and equals $\log N_s$, where $N_s$ is the number of states of entity (i), i.e., the cardinality $|x_{t+1}^{(i)}|$. Given that we are summing over all possible values of the previous time slice ($N_a$ is the number of possible actions and $N_s^{N_e}$ is the number of possible states of all $N_e$ entities in the previous time slice), and over all the entities $N_e$ in the current time slice, the maximum value of the numerator is $N_eN_aN_s^{N_e}\log N_s$, which is used as normalizing denominator. $H_\text{MDP}$ would equal 1 only when the dynamics of the factored MDP are purely random.
>
> In the IPPC case, the problem setup gives us direct access to $p(x_{t+1}^{(i)}|x_t,a_t)$, where the conditioning only depends on a small number of entities from the previous timestep, so it is possible to compute $H(p(x_{t+1}^{(i)}|x_t,a_t))$ exactly for all entities and all $x_t, a_t$ and simply average those (which is equivalent to the summing and normalizing of the definition above for $H_\text{MDP}$).
>
> **About the definition of ${\pmb x}$ and $\pmb a$**
>
> These are defined in Section 2.1 (line 43 to be more precise).
>
> **$H_\text{Bethe}$ is not used in line 118**
>
> You are absolutely correct, we have fixed this. The corrected equation is
>
> $$
>   H_\text{Bethe}^\text{planning}({\pmb{\tilde q}})
>    = \sum_{i=1}^{N_e} H_{\pmb{\tilde q}}(x_1^{(i)})+
> \sum_{t=1}^{T-1} \Big(H_\text{Bethe}( \tilde{q} (\pmb{x}\_t))+
> \sum_{i=1}^{N_e}
>  H_{\pmb{\tilde q}}(x^{(i)}_{t+1}| {\pmb x}^{\text{pa}(i)}_t, a_t)
> \Big)
> $$
>
>
> **About the update equations in Sec 3.3**
> Thanks for pointing this out. Since these equations arise from deterministic distributions, we think the best option is to use a Kronecker delta $\delta_{a,b}$, where $\delta_{a,b} = 1$ if and only if $a = b$ and 0 otherwise. This results in
>
> Forward updates:
> $$
> m_\text{f}(x_1)=P(x_1);~~~m_\text{f}(x_{t+1})=\sum_{x_t,a_t} p(x_{t+1}|x_t,a_t)\delta_{a_t, \arg\max_{a_t'} Q(x_t,a_t') })m_\text{f}(x_t)
> $$
>
>
> Posterior:
> $$
> q(x_{t+1}, x_t, a) \propto m_\text{b}(x_{t+1})p(x_{t+1}|x_t,a_t)\delta_{a_t, \arg\max_{a_t'} Q(x_t,a_t')} m_\text{f}(x_t)
> $$
>
> **About the normalized entropy and missing definitions**
>
> The terms $N_s, N_e$ are introduced in Section 2.1, we will add the definition of $N_a$ (which is the number of actions). The normalized entropy is probably the most naive way to measure the randomness of a factored MDP, and as far as we are aware has never been proposed before. It was very useful to characterize the behavior of the different types of inference.

---

> > ### Comment · Reviewer_QvAf · 2024-08-09
> >
> > I thank the author for their response and clarifications. I have no further questions.

---

### Decision · Program_Chairs · 2024-09-25

**Decision:**

Accept (spotlight)

**Comment:**

The paper proposes a framework for establishing a correspondence between various inference queries in graphical models and planning in MDPs. The paper relies on variational inference approach to inference in graphical models, and unifies inference for planning with other type of queries. The theoretical results are confirmed by empirical evaluation on several domains.

While all or at least most reviewers welcomed the insights provided by the paper, most also pointed at lack of clarity of the presentation, as well as at the lack of rigor in definitions and theoretical analysis. Some of the concerns were addressed in the discussion between the authors and the reviewers, while others, in particular clarity of exposition and rigor of definitions, still stand. The reviewers conditioned their evaluations on improved clarity and less dependence on external background.

Based on the reviews and the discussions, I recommend accepting the paper for spotlight presentation . I believe that spotlight presentation strikes the right balance between exposure of the paper to the audience, and the need for more detailed and interactive discussion of paper details than a full-length oral presentation would provide.